# Mechanical Properties of SLM-Printed Aluminium Alloys: A Review

**DOI:** 10.3390/ma13194301

**Published:** 2020-09-26

**Authors:** Panneer Ponnusamy, Rizwan Abdul Rahman Rashid, Syed Hasan Masood, Dong Ruan, Suresh Palanisamy

**Affiliations:** 1Faculty of Science, Engineering and Technology, Swinburne University of Technology, Hawthorn, VIC 3122, Australia; pponnusamy@swin.edu.au (P.P.); smasood@swin.edu.au (S.H.M.); druan@swin.edu.au (D.R.); spalanisamy@swin.edu.au (S.P.); 2Defence Materials Technology Centre (DMTC) Limited, Hawthorn, VIC 3122, Australia

**Keywords:** selective laser melting, aluminium alloys, mechanical properties, tensile, compression, fatigue, heat treatment, build orientation

## Abstract

Selective laser melting (SLM) is a powder bed fusion type metal additive manufacturing process which is being applied to manufacture highly customised and value-added parts in biomedical, defence, aerospace, and automotive industries. Aluminium alloy is one of the widely used metals in manufacturing parts in SLM in these sectors due to its light weight, high strength, and corrosion resistance properties. Parts used in such applications can be subjected to severe dynamic loadings and high temperature conditions in service. It is important to understand the mechanical response of such products produced by SLM under different loading and operating conditions. This paper presents a comprehensive review of the latest research carried out in understanding the mechanical properties of aluminium alloys processed by SLM under static, dynamic, different build orientations, and heat treatment conditions with the aim of identifying research gaps and future research directions.

## 1. Introduction

Additive manufacturing (AM), more popularly known as 3D printing (3DP), has been extensively applied for various engineering applications. When compared to conventional manufacturing, highly complex parts such as lattice structures can be produced by AM because of the layer-by-layer fabrication process [1]. In recent times, AM technologies are being sought after for providing customized solutions to problems arising due to the COVID-19 virus [2]. Several technologies are in practice for the AM of metallic parts, which include selective laser melting (SLM), electron beam melting (EBM), laser engineered net shaping (LENS), direct metal deposition (DMD), and cold spray additive manufacturing (CSAM). Each of these processes offers its own merits and limitations in terms of quality of the print-part, mechanical property, performance of the component, and the range of materials that can be fabricated. Among these AM processes, SLM is being widely accepted by industry to manufacture customized, high value-added, and complex metal components for aerospace, automotive, defence, and biomedical applications [3]. SLM is a powder-bed fusion type process that produces metal parts by selectively fusing metal powders on a platform using a laser beam. Figure 1 presents the schematic illustration of the SLM process. According to Kempen et al. [4], the fusing occurs through the melting and rapid solidification of metal powders scanned by a laser beam along a 3D print-path created by a processing software. 

Aluminium alloys, titanium alloys, steels including stainless steels and tool steels, nickel superalloys, and cobalt-chromium alloys are the most common SLM-processed metals. There is a growing need for SLM to create fully dense parts, where mechanical, thermal, and other properties are comparable to those of the wrought and/or conventionally fabricated materials. However, SLM is a very complex process and a lot of research is targeted at understanding the effects of various process parameters on mechanical properties, performance, and quality of printed parts. Some of the main process parameters affecting part quality and properties in SLM are the laser power, hatch spacing, defocus distance, powder layer thickness, scan speed, and scan strategy. Maamoun et al. [6] presented a detailed study on the influence of these processing parameters on relative density, porosity, surface roughness, and dimensional accuracy. They identified an optimal processing window using process maps to achieve desired values for each performance characteristic. Likewise, Rashid et al. [7] have reported that varying a single SLM parameter, scan strategy, can result in significantly different microstructural and mechanical properties of the printed parts. Therefore, it is very important to understand the effects of the different SLM process parameters on the part quality and properties.

Aluminium (Al) and its alloys are characterized by their light weight, high strength, corrosion resistance, and good weldability, making them suitable for a range of applications in industries such as automotive, aerospace, machinery and tooling, defence, and construction. Of the different alloy combinations, aluminium-silicon-based alloys (Al-Si), specifically AlSi_10_Mg, AlSi_12_, A356 (AlSi_7_Mg_0.3_), and A357 (AlSi_7_Mg_0.7_), have been extensively used in the SLM process owing to their fabricability. 

This paper presents a comprehensive review on latest research conducted on mechanical properties of Al alloys processed by SLM. A large number of articles have been published on process optimization, anisotropy effects due to build orientation, and the improvement of SLM built part properties by heat treatment. This paper presents a classification and discussion of these published papers into various research categories to identify research directions and research gaps for different types of mechanical properties and performance under static and dynamic loading conditions.

## 2. 3D-Printing of Aluminium Alloys by SLM

Aluminium alloys processed by SLM are attracting attention due to their light weight, high strength to weight ratio, corrosion resistance, and good mechanical properties, and due to the unique advantages offered by SLM, such as tool-less fabrication, geometric freedom, customized design, and intricate shapes. As a result, there is an increased interest from researchers to fabricate Al-Si alloy components using SLM with desirable properties.

Even though Al-Si alloys find extensive applications in the cast form, they are difficult to process using SLM because it is a laser-based process with rapid melting and solidification. Al powders are lightweight, have poor flowability, high reflectivity, and high thermal conductivity. Moreover, Al powders have low laser absorption and are prone to oxidation and balling [8]. Montero-Sistiaga et al. [9] have stated that, besides high reflectivity, high reactivity with oxygen, and high conductivity, there are potential challenges in processing Al alloys using SLM. Despite these drawbacks, aluminium can be alloyed with other metals to overcome some of these challenges. Therefore, the two most common Al-Si alloys used in SLM systems are AlSi_10_Mg and AlSi_12_. The addition of silicon to aluminum improves its fluidity and reduces its melting temperature. The addition of magnesium to aluminum increases its strength through solid solution strengthening and also improves its strain hardening ability [10]. Although significant amount of works are published on AlSi_10_Mg, the research work on AlSi_12_ and other Al-Si alloys, including A356 and A357, is still evolving [11,12,13,14,15,16,17,18,19,20,21,22,23,24,25,26].

Table 1 presents a compilation of key published literature on the SLM of these Al alloys, which are reviewed in detail in this study, capturing the main findings of each article, considering the process conditions (build orientation, heat treatment, and process optimisation), types of mechanical properties, and type of SLM machine used. It was also noted that most of the researchers have used commercial SLM machines produced by leading SLM manufacturers, such as SLM Solutions, EOS, Concept Laser, Realizer, and Renishaw. More detailed discussion on some of these publications is presented in later sections of this review. 

It is evident from the information presented in Table 1 that there is a lot of interest in understanding the SLM processability of various aluminium alloys. Olakanmi et al. [139] reported in detail the influence of process parameters, powder properties, and laser types on the densification of aluminium alloys. Moreover, they also reported the potential issues arising during the SLM of aluminium alloys, such as metallurgical defects, porosity, and oxidation. The primary causes for these processing challenges include high reflectivity, high thermal conductivity, and poor flowability of aluminium alloy powders, which, as reported by Aversa et al. [140], can be addressed by the development of new alloys specifically designed to improve the SLM-printability of various aluminium alloys. This idea was corroborated by Sercombe and Li [141] who reported improved microstructures and enhanced mechanical properties of various aluminium metal matrix composites fabricated via SLM.

The primary research focus areas of most of the published literature on SLM of aluminium alloys can be broadly classified into four categories, viz, process parameter optimization, effects of build orientation, in-situ and post heat treatment strategies, and modelling and numerical analysis of print-part performance as listed in the overview of publications in Table 2. It is clear from this overview that AlSi_10_Mg alloy has received maximum attention of research papers for all four research focus areas for mechanical properties, with a large number of researchers paying attention to the effects of build orientation and heat treatment of this alloy. More research needs to be done on process optimization and effects of the build orientation and heat treatment of AlSi_12_ alloy and other two less popular A356 and A357 alloys processed by SLM. As reported by Rometsch et al. [142] and Zhang et al. [143], these studies aim to improve the performance of SLM-printed aluminium components via: (a) optimizing powder characteristics, (b) optimizing process parameters to obtain 100% dense parts, (c) minimizing the defects such as residual stresses, distortion, and cracking, (d) optimizing post heat treatment techniques to achieve desired mechanical properties, (e) understanding the microstructural characteristics of the print-part, and (f) anisotropic mechanical properties as a result of varying build orientation.

## 3. Mechanical Properties of SLM-Printed Aluminium Alloys

Characterisation of the mechanical properties of any additively manufactured part is generally very demanding. Although ASTM F3122-14 presents standard guidelines to evaluate most of the mechanical properties of 3D-printed metallic materials, there is no set standard developed for this purpose, making it difficult to benchmark and/or compare the properties of the same material printed using different AM printers with varying geometries. Therefore, extensive research has been published to characterize various types of mechanical properties of SLM processed Al alloys under static, dynamic, and heat treatment conditions. Table 3 is a compilation of the various mechanical properties investigated by researchers for four main types of SLM-printed aluminium alloys, namely AlSi_10_Mg, AlSi_12_, A356, and A357.

Table 3 shows that maximum research effort has been directed to investigate the tensile properties of SLM processed aluminium alloys. In most of these publications, effect of build orientation and heat treatment on microhardness, yield strength (YS) and ultimate tensile strength (UTS) has been studied. Many of these publications have also discussed the microstructure variation in the samples arising due to rapid heating and cooling encountered in the SLM process and relate them to the mechanical property relationships. Table 3 also reveals that some important mechanical properties such as quasi-static compression, dynamic (high strain rate), fatigue, impact, flexural and wear have received very little attention of researchers. These properties are of significance in many automotive, aerospace, and biomedical applications.

Heat treatment is one of the essential processes to achieve the desired property such as the ductility of the part produced by SLM. Ma et al. [148] have applied the heat treatment of hyper-eutectic alloy AlSi20 to investigate the mechanical properties as well as microstructure. In the heat treatment process, Si particles became coarser and the morphology changed from fibrous to plate like structure. Also, as anticipated, the changes in the microstructure during heat treatment affects the mechanical properties. A significant decrease in strength and, on the other hand, a very high increase in ductility were observed.

A detailed description of research carried out on each of the above-mentioned aluminium alloys is discussed below.

### 3.1. SLM-Printed AlSi_10_Mg

AlSi_10_Mg is one of the most common aluminium alloys and finds wide applications in various industries of 3D printing applications such as automotive and aerospace. The effect of build orientations in horizontal (H), inclined (I), and vertical (V) directions, and the effect of post heat treatment on the tensile mechanical properties of SLM-printed AlSi_10_Mg are summarized in Table 4. In most of the studies, the tensile samples built in horizontal direction on the build plate seem to provide higher tensile strength than the sample build in vertical direction. In most studies, it was observed that heat treated (HT) SLM samples display reduced tensile strength and increased ductility than the as-built (AB) SLM samples. Very few studies have been conducted to assess the effect of build orientation on heat treated SLM samples. Table 4 also presents results of some studies carried out on fatigue and fracture toughness of the SLM-built parts. Very few studies have been undertaken on flexural behaviour of the printed aluminium parts. Table 4 shows results of work carried out on the quasi-static and dynamic compression of SLM processed aluminium samples. Again, a great deal of studies are required to understand the quasi-static compression and high strain rate dynamic tensile and compressive behaviour of SLM-processed aluminium parts for different orientations and heat treatment conditions. Moreover, the mechanical performance of SLM-processed aluminium alloys operating at elevated temperatures have also not received the attention of researchers.

According to Aboulkhair et al. [149], when the process parameters and scan strategy were optimised, it would affect the density of AlSi_10_Mg parts produced by SLM directly. Therefore, the porosity component was evaluated, and the part was found to be 99.8% dense. In a study by Read et al. [108], the mechanical properties of the SLM built AlSi_10_Mg samples were found to perform better than as-cast alloys of the same composition. The mechanical properties tested were tensile strength, creep, and the porosity. Similarly, Li et al. [80] reported that the solution treatment of as-built AlSi_10_Mg samples produced by SLM yielded better ductility (from 5% to 24% approximately), while the tensile strength was significantly reduced from 434 MPa to 168 MPa approximately. Li et al. [81] investigated the as-built SLM AlSi_10_Mg samples and tested them at −70 °C for mechanical properties. The fish-scale morphology along the build direction and oval structures on the direction perpendicular to the build direction were observed.

A study by Hitzler et al. [68] focused on the anisotropic compressive behaviour of SLM AlSi_10_Mg. They found that compressive Young’s modulus was significantly higher than the Young’s modulus of tensile loading as well as Young’s modulus of bulk base material. The compressive Young’s modulus found to be about 82 GPa. Furthermore, they reported that the compressive yield strength was found to be similar to tensile yield strength, however the ultimate compressive strength was obtained to be significantly higher than the tensile strength.

Wang et al. [150] studied the effect of energy density on densification behaviour and surface roughness of SLM AlSi_10_Mg. The densification behaviour was analysed using X-ray and CT scanning. High relative density was obtained through the point distance of 80–105 µm with an exposure time of 140–160 µs. Furthermore, energy density was stated to have a significant influence on the surface morphology. On one hand, the increased energy density could lead to a balling defect and, on the other hand, reduced energy density could cause defects, such as porosity and micro-cracks.

Everitt et al. [54] investigated the hardness through nano-indentation and found that AlSi_10_Mg samples produced by SLM have higher hardness than cast counterpart. Furthermore, the microstructural studies revealed that grain sizes were increased at the melt pool edges, which were associated with the homogenous distribution of Si particles throughout the sample.

A study by Leon and Aghion [151] evaluated the effect of surface roughness on SLM processed AlSi_10_Mg followed by stress-relief heat treatment. They focused on corrosion resistance and corrosion fatigue performance. Furthermore, the results of SLM processed AlSi_10_Mg samples were compared with the cast counterparts. Polishing was found to improve the corrosion fatigue life span of SLM samples, whereas the unpolished SLM samples caused an increase in surface roughness and other surface defects.

Ch et al. [43] studied the SLM AlSi_10_Mg alloy under two different environments namely argon and nitrogen, while building the parts in both horizontal and vertical orientations. Their microstructural findings include cellular structure showing α-Al matrix displaying Al and Si eutectic mixture. Tensile strengths were found to be 385 MPa and 338 MPa for the nitrogen and argon atmosphere respectively. Furthermore, the samples built in the horizontal orientation showed a variation of about 5% in the nitrogen environment, whereas in the vertical orientation, 7.5% variation of strength was found. Based on their findings, nitrogen was inferred to have preference over argon as the shielding gas.

Takata et al. [121] studied the microstructure and mechanical properties of SLM processed AlSi_10_Mg samples heat treated at temperatures 300 °C and 500 °C. They reported that, at elevated temperatures, fine Si phase precipitation occurred resulting in coarse eutectic Si particles. The as-built part was found to exhibit a tensile strength of 480 MPa. Furthermore, the tensile strength was reported to be isotropic, whereas the tensile ductility was found to be anisotropic. Even, the anisotropic property was found to disappear at a heat-treated temperature of 530 °C.

Girelli et al. [152] investigated the effect of temperature, solution treatment, and ageing on the microstructure, microhardness, and density of SLM AlSi_10_Mg. They also investigated the AlSi_10_Mg samples produced by gravity casting under the same heat treatment conditions. The as-built SLM samples were said to exhibit superior mechanical properties than gravity casted samples due to the refinement of grains and nano-sized Si particles. The as-built AlSi_10_Mg samples had a fine microstructure and the heat treatment was found to decrease the ultimate tensile strength but not the yield strength.

Liu et al. [85] investigated the as-built AlSi_10_Mg samples produced by SLM and reported the issues with respect to surface roughness and mechanical properties. Sandblasting had shown a difference of 80% in surface roughness values. Furthermore, the results of tensile tests revealed superior outcome than the die-cast counterparts. A similar behaviour was found in terms of hardness and density.

In a study by Raus et al. [107], mechanical properties namely microhardness, tensile strength and impact toughness were investigated. The results were then compared with conventional high pressure die cast A360 alloy. The results revealed that microhardness and yield strength were higher for the SLM AlSi_10_Mg by 42% and 31% respectively than the die cast counterparts. They used the SLM process parameters as laser power of 350 W, scanning speed of 1650 mm/s, and hatching distance of 0.13 mm to attain the part density of 99.13%.

Recently, Hadadzadeh et al. [61] studied the strain rate behaviour for AlSi_10_Mg alloy processed by SLM under dynamic loading in two build orientations, one horizontal and the other being vertical. The strain rate was said to increase rapidly in both the cases, while the strain rate at vertical orientation was ~1400 s^−1^. Both the flow stresses follow a similar trend, whereas the flow stress in the vertical orientation were stated to progress further with increase in strain.

Nurel et al. [95] have also reported the dynamic properties of SLM processed AlSi_10_Mg at strain rates of 7 × 10^2^–8 × 10^3^ s^−1^ using SHPB. The SLM AlSi_10_Mg was found to have been affected by build orientation and heat treatment. They also investigated the anisotropic nature of SLM parts using ellipticity after the SHPB tests, but they have not observed any strain rate sensitivity. The focus of this research was on the significance of the build orientation at different strain rates. No significant correlation of build orientation with respect to strain rate was found in the range of 1000–3000 s^−1^.

Asgari et al. [31] have also investigated the dynamic behaviour and texture of AlSi_10_Mg with build plate heating at 200 °C. Their findings showed that, in vertically built SLM samples, shock loading did not have an influence on texture and in horizontally built samples, transition of texture was observed (at a strain rate of 1600 s^−1^). The dynamic mechanical behaviour was investigated using split Hopkinson pressure bar at a strain rate ranging from 150 s^−1^ to 1600 s^−1^.

Trevisan et al. [10] reviewed the process, microstructure and mechanical properties of SLM AlSi_10_Mg. The focus was on mechanical properties coupled with very fine microstructure, which is primarily due to rapid melting of AlSi_10_Mg powder and fast solidification. The review analysed the effect of main process parameters, namely laser power, scan speed, layer thickness, and scan strategy. In addition, the effect of heat treatment on tensile as well as fatigue properties associated with the SLM AlSi_10_Mg samples were discussed.

### 3.2. SLM-Printed AlSi_12_

The AlSi_12_ alloy is increasingly gaining focus because it is a near eutectic alloy characterised by good casting ability and high specific strength. The low melting point of this alloy provides good castability due to its low density and good wear properties, AlSi_12_ is quite attractive for applications in automotive and aerospace industries. It can be used for cryogenic applications as it can retain its strength at low temperatures such as at high altitudes [106]. The mechanical properties of SLM-printed AlSi_12_ are summarized in Table 5. It is noted that SLM processed AlSi_12_ alloy has received less attention for mechanical characterization compared to AlSi_10_Mg alloy in terms of heat treatment and build orientation effects. Most of the comments mentioned earlier concerning the need for further research on different properties mentioned for the AlSi_10_Mg alloy also apply for SLM printed AlSi_12_ alloy.

Siddique [14,115] reported that in automotive and aerospace industries, there is trend to replace cast iron with Al-Si alloys to achieve light weighting in components such as engine blocks, cylinder heads, pistons, power trains, and intake manifolds. Many of these parts are moving and operating under dynamic loading conditions with elevated temperatures. Therefore, extensive work needs to be done to understand the mechanical behaviour of Al-Si alloys processed by SLM under dynamic loading, impact, wear, and elevated temperature conditions.

Prashanth et al. [17] investigated mechanical behaviour of SLM AlSi_12_ and reported that the yield strength and tensile strength were 260 MPa and 380 MPa respectively. The effect of annealing on the microstructure and on the tensile properties revealed that mechanical properties could be fine-tuned, and wide range of strength and ductility could be achieved. In another study, Prashanth et al. [16] evaluated the effect of annealing on tribological and corrosion properties of SLM processed AlSi_10_Mg through sliding and fretting wear tests. The sliding wear showed less wear rate when compared to the cast parts. The studies revealed that wear properties and corrosion behaviours have a correlation with each other, as both are strongly associated with a change in microstructure. The change in microstructure was due to the heat treatment, as the Si particles grow, and the density decreases with increasing annealing temperature. 

In another investigation, Prashanth et al. [153] carried out compression tests on both AlSi_12_– Titanium, Niobium, and Molybdenum (TNM) composites and AlSi_12_ matrix produced by SLM and studied their behaviour. The compression tests revealed that TNM composites–AlSi_12_ matrix had significantly better compressive strength when compared to the AlSi_12_ matrix. However, the TNM composites had revealed less plasticity. Prashanth et al. [101] also analysed the tensile behaviour of SLM AlSi_12_, where they applied the base plate heating and four different hatch styles. The samples showed similar crystallite sizes but different textures. In addition, the samples produced with base plate heating had lesser residual stress. The results reported were the yield strength ranging between 235 MPa and 290 MPa, while the ultimate tensile strength varied between 385 MPa and 460 MPa. Furthermore, the ductility varied between 2.8% to 4.5%. Since the base plate heating produced desirable results, they carried out base plate heating at three different temperatures namely 473 K, 573 K, and 673 K, which yielded the tensile ductility of 3.5%, ~3%, and 9.5% respectively.

Siddique et al. [116] used the computer tomography for investigating the fatigue performance of SLM AlSi_12_ samples. The samples revealed better or at least comparable yield strength or ultimate tensile strength when compared to cast parts. Surface roughness was found to affect the fatigue strength, which was improved by post processing. Furthermore, a porosity study was essential to employ the samples for high cycle applications. The computer tomography was applied to study the stress concentration due to porosity. In another study Siddique et al. [15] determined the effect of process parameters and post processing on the microstructure and mechanical properties of SLM processed AlSi_12_. They performed quasi-static tests and fatigue tests for mechanical characterisation. Furthermore, they observed extra-ordinary eutectic microstructure using scanning electron microscope. By changing the build rate and keeping the process in control, mechanical properties obtained were found to be better for SLM parts than cast counterparts. Moreover, Siddique et al. [14] focused on the design considerations and mechanical properties of SLM processed AlSi_12_ parts. From the application perspective, the parts were designed to perform in the high cycle environment. Therefore, the study was more focused on very high cycle fatigue and the corresponding crack propagation behaviour in the cyclic environment. They further investigated the base plate heating and post processing, especially stress relieving, which are critical for the fatigue loading of AlSi_12_ parts. The base plate heating was found to influence the fatigue crack initiation mechanisms.

Wang et al. [131] studied the effect of build chamber atmosphere on the SLM AlSi_12_ samples. The effect of three different atmosphere namely argon, nitrogen and helium were studied for their influence on the mechanical properties. Hardness, density, and relative density were analysed and found to have similar attributes for all the atmospheres. However, the mechanical properties of parts produced in argon and nitrogen atmospheres were found to be better than the helium atmosphere. Furthermore, the mechanical properties in SLM parts were better than those produced by conventionally made AlSi_12_ counterparts.

Lykov and Baitimerov [154] investigated the SLM AlSi_12_ to determine the process parameters to ensure the least porosity. The samples were found to have varied microstructures and the porosity was reported to be about 0.5%. The AlSi_12_ powders were found to have poor flowability. The surface morphology revealed that samples have an uneven surface. They recommended that to reduce the surface roughness and porosity, it was essential to use AlSi_12_ powder with fine fraction of about 17 µm and the scanning strategy include double pass laser scan strategy.

Mendřický and Keller [155] focused on the precision, geometry and dimensional accuracy of the AlSi_12_ parts produced by SLM. The part orientation was planned to minimise the internal stress. The SLM part was digitised to collect the sample data. The collected data was compared with the CAD data to evaluate for dimensional and geometrical accuracy. The deformation of the model was caused by the internal stress on the as-built AlSi_12_ parts.

Šafka et al. [156] analysed the build orientation and the placement of part in the SLM build substrate, which could reduce the amount of support structures required and improve the flexural strength of SLM AlSi_12_ parts. They observed that the least energy and therefore less heat would lead to little deformation of the printed final part.

Chou et al. [45] proposed a new approach of using pulsed SLM over the conventional SLM for AlSi_12_ and achieved greater control of the heat input. In the SLM AlSi_12_, Si refinement below 200 nm was attained. Furthermore, 95% of dense parts and the hardness of 135 HV were achieved. The part produced by pulsed SLM method yielded better hardness than the conventional cast alloy counterpart. The cooling rate and thermal gradient had an influence on solidification phase and the microstructure.

In a study by Vora et al. [13], a novel method had been discussed to produce aluminium parts with the anchorless SLM. This anchorless method is opposed to the regular method that uses anchor or support for the SLM parts. Furthermore, the anchorless SLM (ASLM) could produce SLM parts that can remain in the stress-free state. The ASLM was found to be suitable for processing eutectic alloys, as well as hypo eutectic and hyper eutectic alloys. In addition, they studied the in-situ alloy formation within SLM chamber. The residual stress was also found to be less with the ASLM processed AlSi_12_ alloy.

Suryawanshi [120] investigated the tensile strength and toughness of SLM AlSi_12_ and found that mechanical properties were relatively better than the cast counterparts, except the ductility. The scan strategy especially linear vs checker-board hatch style was analysed and the later was found to have significant effect on the tensile strength. Furthermore, the meso-structure due to the laser hatch tracking leads to improved fracture toughness. In addition, to attain higher fatigue strength, the residual stresses, porosity, and un-melted powder particles needed to be removed. The SLM AlSi_12_ was found to give more avenues for material design and fabrication with enhanced strength and toughness.

In a study by Li et al. [82], the SLM AlSi_12_ was reported to have controllable and ultrafine microstructure. The excellent mechanical properties were achieved through solution heat treatment. They introduced a novel approach for refining the eutectic alloy of AlSi_12_, which yielded better tensile properties than cast counterparts. The solution treated AlSi_12_ was found to produce the tensile ductility of ~25%. Furthermore, the tailoring of the mechanical properties according to the applications was possible by controlling the solution heat treatment time.

Louvis et al. [8] investigated the process parameters of SLM AlSi_12_ to achieve uniform relative density. The process parameters investigated were laser power and the laser scanning rate. The investigation was extended to understand the difficulty of processing AlSi_12_ when compared to stainless steel and titanium alloy. The major factor that affected the relative density was the oxidation factor. The process parameters were varied using two different SLM machines, one with 50 W laser power and the other with 100 W laser power. Even with the optimum process parameters combination, the machine with 100 W laser power yielded only the relative density of 89.5%. Furthermore, it was recommended that to produce SLM AlSi_12_ parts with 100% relative density, it was essential to develop methods to disrupt the oxide formation or to prevent oxidation.

Ackermann et al. [157] studied the fabrication of thin structures and their applications using SLM AlSi_12_ parts. In service, the applications of SLM AlSi_12_ parts were found on microelectronics, fine mechanical structures, and automotive mechanisms. The powder characteristics such as mean particle size and powder shape, the process parameters such as laser spot size and scanning speed were found to influence the thickness of fine structures. They arrived at 0.21 mm as the optimum thickness for thin structures, for sufficient strength.

Rathod et al. [106] investigated the effect of scanning strategy and heat treatment on the tribological properties of the SLM AlSi_12_ parts and compared with cast alloy. The process of annealing lead to Si precipitation resulting in the reduction of hardness. The as-built AlSi_12_ SLM samples were reported to have less wear rate when compared to heat treated SLM samples and cast samples.

### 3.3. SLM-Printed A356 and A357

There are two other types of Al-Si alloys, A356 and A357, which have been processed by SLM. These two alloys are also Al-Si-Mg alloys. They both contain 7%Si by weight, but they have a slightly different Mg content. The alloy A356 is AlSi_7_Mg_0.3_ and alloy A357 is AlSi_7_Mg_0.7_. Very few studies have been made on mechanical characterisation of these two alloys processed by SLM. The mechanical properties of SLM-printed A356 and A357 are summarized in Table 6.

Kimura and Nakamoto [20] optimised process parameters to arrive at dense SLM A356 (AlSi_7_Mg_0.3_). The relative density attained was 99.8% with the laser irradiation conditions. The mechanical test results found the ultimate tensile strength of ~400 MPa, the yield strength of ~200 MPa and the elongation from 12–17%. The heat treatment revealed the difference in microstructure and mechanical properties of SLM A356 samples and as-cast samples. After annealing, the SLM samples became elongated to 30%, while the tensile strength of A356 samples reduced by half the value and became ~200 MPa.

Rao et al. [25] investigated the mechanical properties and microstructure of SLM A357. The processing parameters were optimised to achieve dense A357 samples, along with fine microstructure. The porosity was also analysed based on relative density and laser parameters. The anisotropy of SLM A357 Al alloy samples were investigated in the horizontal and vertically built tensile samples. Fractographic studies revealed that horizontal orientation was better for printing the A357 tensile samples. In another study Rao et al. [24], the tensile behaviour of SLM A357 in the as-built and heat treated condition was investigated. The as-built sample displayed ultrafine microstructure. The tensile samples of SLM A357 were found to have better properties than their cast counterparts. The Al grains in the as-built parts had eutectic nano-sized Si particles, contributing for higher strength. However, the nano-sized Si particles did not favour the ductility. After the heat treatment of SLM A357, as anticipated, the tensile ductility improved and was reported to be about 23% with reduced tensile strength.

Yang et al. [26] investigated the effect of heat treatment on SLM A357 by focusing on stress relief to analyse the mechanical property and microstructure. For the as-built A357 samples, the rapid melting and rapid cooling affects the intermetallic phases such as Mg_2_Si precipitates. During the heat treatment process, the breaking up of the Si network was said to occur, which lead to high ductility. Si particles at grain boundaries were observed to coarsen. Furthermore, anisotropy was observed to disappear in terms of affecting yield strength and ductility when the microstructure became more homogenised.

### 3.4. Other Aluminium Alloys Processed Using SLM

From the research perspective and for the applications with desired mechanical properties, new Aluminium alloy are being developed and processed using SLM. These new aluminium alloys have been developed by alloying them with new elements, such as Cu, Ni, Sc, Zr, V, and Zn, to impart improved properties and better processing by SLM.

Aversa et al. [158] developed Al-Si-Ni alloy samples for SLM with a combination of AlSi_10_Mg and Ni powders. This combination was found to be near eutectic by composition. Furthermore, this combination of AlSi_10_Mg and Ni were reported to have better hardness than AlSi_10_Mg. The increase in hardness was mainly attributed to the addition of Ni, which was demonstrated through nano-indentation measurements. The SLM process parameters were optimised to reduce porosity. The results obtained with the combination of AlSi_10_Mg and Ni had also yielded Al3Ni agglomerates.

Spierings et al. [159] developed Sc and Zr-modified AlMg alloys for SLM processing. The Sc and Zr-modified Al alloy is commercially known as “Scalmalloy” and offers more advantages over traditional 4xxx as-cast alloys. These advantages include high strength and high ductility at low anisotropy. Furthermore, fine grain microstructure and weak texture along the build direction causes low anisotropy. They also analysed the influence of larger scan speeds on the mechanical properties of Scalmalloy and found that grain sizes were reduced from 1.1 µm to 600 nm.

Zhang et al. [160] investigated the manufacturing of Al-Cu-Mg alloys using SLM. They studied the effect of process parameters on the density of SLM Al-Cu-Mg alloys. The energy density was found to have significant influence on the densification of Al-Cu-Mg alloys. The threshold value of energy density was 340 J/mm^3^, that yielded the sample density of 99.8%. Furthermore, fine microstructure was obtained without any imperfections and micro-cracks. A high value of ultimate tensile strength of ~400 MPa and the yield strength of ~275 MPa were obtained for the SLM Al alloy part. In addition, they found fine grains and solid solution strengthening mechanism for the higher mechanical strength. In another study, Zhang et al. [161] investigated the microstructure and mechanical behaviour of SLM Al-Cu-Mg alloy and Zirconium modified SLM Al-Cu-Mg alloys. The addition of Zr had the effect of reducing the hot cracking in the SLM part. The comparison of Zr modified Al-Cu-Mg alloy and Al-Cu-Mg alloy was found to show an ultrafine grain with Zr addition. Furthermore, Zr modified Al alloy had significantly higher yield strength close to its ultimate tensile strength of ~450 MPa.

Nie et al. [162] studied the effect of Zr on the formability, mechanical properties and microstructure of SLM Al-Cu-Mg-Mn alloys. The addition of Zr to SLM Al-Cu-Mg-Mn alloy was observed to be effective in crack controlling mechanism, grain refinement and in the controlling of mechanical properties. The addition of Zr was found to cause the transformation of grain type from columnar to equiaxed type. Furthermore, they investigated crack inhibition and enhancement of mechanical properties due to the addition of Zr with SLM Al alloy.

Karg et al. [163] investigated the laser beam melting of EN AW-2219 and observed that laser processing of EN AW-2219 (Al-Cu alloys) was very challenging. Porosity and tensile tests were conducted for the SLM Al-Cu alloy. Then, T6 heat treatment was applied to some built samples and the elongation was observed. The elongation was found to exceed in the build direction by a factor of two when compared to traditional cast counterpart.

Martin et al. [164] stated that several types of aluminium alloys cannot be processed using SLM due to the complex melting and solidification dynamics. The issues were found to be microstructures with larger columnar grains and periodic cracks. However, they suggested methods to resolve these issues by introducing nanoparticles that would control the solidification. Then the powder particles were found to be qualified for SLM. Furthermore, the technique of solidification control could be applied for the conventional process as well, where hot cracking and hot tearing occurs commonly.

Montero-Sistiaga et al. [9] studied the SLM Al7075 (Al-Zn-Cu-Mg), a wrought alloy, to improve its density. The SLM Al7075 initially displayed poor density and micro-cracking, but it was improved by adding 4% Si. Heat treatment was carried out to improve the hardness of SLM Al7075. Furthermore, the focus was more on producing high strength aluminium alloys.

Aversa et al. [165] investigated the mechanical, metallurgical and the thermal properties for SLM of Al-Si-Zn-Mg-Cu alloy. The introduction of Si was analysed with respect to Al-Zn-Mg-Cu alloy and the crack density was found to be reduced. This crack density reduction was due to improved molten phase fluidity and the reduction of the co-efficient of thermal expansion. Furthermore, the SLM Al-Si-Zn-Mg-Cu alloy was found to be promising in terms of microstructure, microhardness, and the tensile properties.

Zheng et al. [166] reported the variation of microstructure and hardness of an SLM Al-8.5Fe-1.3V-1.7Si. Among the melt pool, three different zones namely laser melted zone, melting pool border and heat affected zone were identified. Microhardness results of the SLM process exceeded the as-cast counterpart. Furthermore, with the decrease in laser scanning speed, laser melted zone was found to reduce significantly.

Croteau et al. [167] investigated the microstructure and mechanical properties of SLM Al-Mg-Zr alloys. The energy densities applied for the fabrication of SLM parts ranged between 123 and 247 J/mm^3^ and the resultant relative density was verified using X-ray tomography. The alloying elements have a significant role, where Mg acted as a solid solution strengthener and Zr contributed towards metastable precipitates resulting in grain refinement and prevention of hot tearing. Furthermore, Zr alloy had improved the mechanical properties.

In a study by Maamoun et al. [88], the influence of process parameters of SLM AA 6061 and AlSi_10_Mg were investigated, based on the relation between microstructure and mechanical properties. AA 6061 is also an Al-Si-Mg alloy with a higher coefficient of thermal expansion (CTE). The process optimisation was carried out on both the aluminium alloys to reduce the defect in microstructure. The mechanical behaviour of the aluminium alloys was analysed through the design of experiments and the results were presented for hardness, yield strength and ultimate tensile strength. The results obtained was useful in improving the part quality that had reduced the post processing requirements.

Uddin et al. [168] analysed AA 6061 using high temperature heating in an SLM environment to yield crack-free parts. The hardness was found to be 54 HV, the yield strength as 60 MPa, ultimate tensile strength of 130 MPa and the elongation of 15%. Furthermore, they revealed that AA 6061 could be successfully manufactured with SLM without displaying the cracking phenomenon.

## 4. Summary and Research Gaps

Although aluminium can be with Zn, Cu, Mg, Mn, and Si to produce age-hardening alloys, casting alloys, and work-hardening alloys, the area of focus in this review was the Al-Si alloys processed by selective laser melting (SLM) metal additive manufacturing technique and their mechanical properties. The Al-Si alloy has the cast-ability and weldability, which forms the unique combination and a suitable candidate for processing in SLM. As mentioned, the most common SLM aluminium alloys are AlSi_10_Mg and AlSi_12_ based on the literature review. This review has focused on classification of published research on SLM-processed aluminium alloys in terms of research carried out on the effects of process optimisation, build orientation and heat treatment on four main types of Al-Si aluminium alloys. The published papers have also been categorised for various types of mechanical properties considered in SLM parts to identify research gaps on which mechanical properties needs more in-depth investigation from the point of view of industrial applications. 

This review has revealed that even though many studies are reported for the tensile mechanical properties of SLM aluminium alloys, very few studies were carried out on the dynamic behaviour in tension and compression, fatigue, impact, wear, and flexural response of the printed parts. Specially very few studies were published on high strain rate loading behaviour of parts made by any metal based additive manufacturing processes like DMD, EBM, and SLM.

One of the other research gaps found in this review is the effect of build orientations on the quasi-static and dynamic compression properties of SLM-processed various other aluminium alloys. It was also observed that some important mechanical properties such as fatigue, impact, flexural, bending, and wear have received very little attention. Moreover, the mechanical performance of SLM-processed aluminium alloys operating at elevated temperatures has also not received much attention. These properties are of significance in many automotive, aerospace, and biomedical applications.

Another important research avenue is to understand the effect of machine-specific characteristics on the properties of the printed components. It was found in this review that researchers have used different SLM machines supplied by various machine manufacturers globally. It is interesting that the same material when printed using different machines yield different mechanical and microstructural characteristics. This can be attributed to the differences in the laser systems used, inherent characteristics of the machine itself, and various other parameters which are not quite understood. Moreover, the repeatability of the mechanical performance of printed parts using same and/or different machines is to be further explored.

Lastly, the post heat treatment plays a crucial role in tuning the microstructural characteristics of the printed parts and render them suitable for desired applications. Most researchers have investigated the heat treatment protocol typically used for conventionally fabricated aluminium alloys, which may not be ideal for SLM-printed parts, as they have different inherent properties. Therefore, further research is required to be carried out to assess the effects of different heat treatment strategies on the performance of printed parts.

## Figures and Tables

**Figure 1 materials-13-04301-f001:**
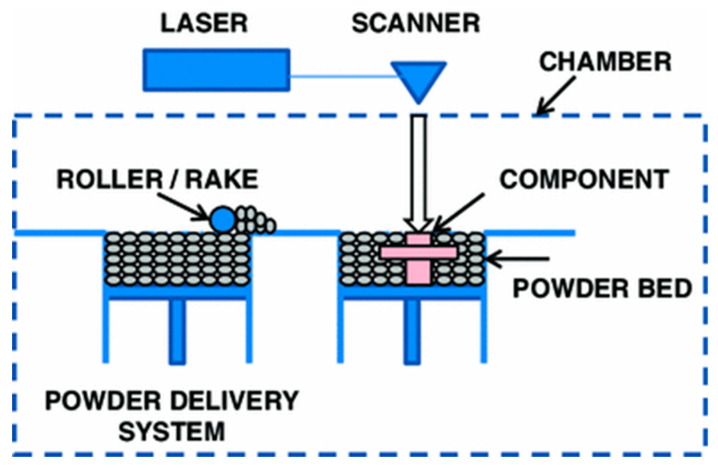
Illustration of the SLM process [5].

**Table 1 materials-13-04301-t001:** List of published literature on SLM of aluminium alloys.

Author (Year)	Material	Process Conditions	Mechanical Properties Evaluated	Main Findings	SLM Machine	Reference
Aboulkhair et al. (2016)	AlSi_10_Mg	Laser power 200 W, layer thickness 25 µm, scan speed 550 mm/s, hatch spacing 80 µm, and scan strategy chess board. Heat treatment: Solution treated at 520 °C for 1 h, water quenched to room temperature and then aged for 6 h at 160 °C.	Fatigue	At lower stress levels, machining the samples improved the fatigue performance, but did not have any influence at higher stress levels. Heat treating the samples significantly improved the fatigue life, and it was found that at 94 MPa stress loading, the heat-treated samples outperformed their cast counterparts in terms of fatigue life. The samples that were heat-treated and machined showed the best fatigue performance.	Renishaw AM250	[27]
Aboulkhair et al. (2016)	AlSi_10_Mg	Laser power 200 W, scan speed 550 mm/s, hatch spacing 130 µm, layer thickness 25 µm, and checkerboard scan strategy. Heat treatment: T6	Tensile strength, compressive yield strength, nano-hardness	The printed parts had tensile strength better than the diecast parts. After heat treatment, the hardness and the tensile strength of the printed parts reduced by about 20% and 12%, respectively, whereas the ductility increased by a factor of 2.8. The compressive yield strength of the heat-treated parts was 169 ± 6 MPa which was approximately half the strength of the as-built parts.	Renishaw AM250	[11]
Aboulkhair et al. (2015)	AlSi_10_Mg	Laser power 200 W, point distance 80 µm, exposure time 140 µs, hatch spacing 130 µm, layer thickness 25 µm and checkerboard scan strategy.	Tensile and compressive strength, nano-hardness	Uniform nano-hardness of 1.82 ± 0.01 GPa was observed in the SLMed samples due to the homogeneous distribution of Si particles segregating at α-Al grain boundaries. Microhardness along the plane parallel to the build plane was found to be higher (109.7 ± 0.9 HV) than the plane perpendicular to the build plane (99.07 ± 2 HV).	Renishaw AM250	[12]
Alghamdi and Haghshenas (2019)	AlSi_10_Mg	Two 400 W lasers at CalRAM Inc. were used, with default SLM 280 parameter set. Heat Treatment: T6, solutionising at 520 °C for 1 h, followed by water quench, then artificial ageing at 170 °C for 4 h.	Nano-hardness	The nano-hardness of the heat-treated samples (1.56 ± 0.11 GPa) was reduced by 42% compared to the strength of the as-built samples (2.71 ± 0.12 GPa). Heat treatment of the samples resulted in the change in microstructure from cellular grains to fragmented/spheroidised eutectic silicon particles. The effect of the indentation size was observed in both as-built and heat-treated samples. With the decrease in the depth of indent, the hardness increased.	400 W laser at CalRAM Inc. SLM 280	[28]
Amani et al. (2018)	AlSi_10_Mg	Laser power 250 W, scanning speed 571 mm/s, layer thickness 60 µm, and argon gas atmosphere.	Compression behaviour	Deformation behaviour of two face-centred cubic lattice structures with thin and thick struts was studied using both in-situ and ex-situ X-ray computed tomography technique. A finite element model was developed using Gurson-Tvergaard-Needleman (GTN) porous plasticity model. Simulations showed a fairly good agreement with the model’s prediction of fracture location.	SLM 250	[29]
Anwar & Pham (2017)	AlSi_10_Mg	Laser power 350 W, layer thickness 100 µm, laser speed 900 mm/s, and hatch spacing of 0.12 mm.	Tensile strength	The quality of the as-built samples depended upon the velocity of the gas flow in the built chamber, the direction of laser scan with respect to the gas flow direction, and position of the part with respect to the chamber outlet location. Higher ultimate tensile strength (UTS) values for parts printed with laser scan direction against the gas flow, with higher gas velocity, and closer to the outlet was observed. When the laser scanning was in the direction of gas flow, a reduced accumulation of spatter powder particles was observed near the outlet.	SLM Solutions 280 HL	[30]
Asgari et al. (2018)	AlSi_10_Mg	Laser power 370 W, scanning speed 1300 mm/s, hatch distance 190 µm, powder layer thickness 30 µm, build platform pre-heating temperature 200 °C. Build orientation: Vertical and Horizontal.	Dynamic compression, Split Hopkinson pressure bar testing	Increasing the strain rate from 150 s^−1^ to 1600 s^−1^ resulted in higher yield, peak flow stress, and ductility. The texture evolution could be attributed to vertical build orientation than horizontal build orientation due to uniform and dense microstructure, while the texture is not affected by the deformation at high strain rates.	EOS M290 machine	[31]
Aversa et al. (2017)	A357	Laser power 195 W, scan speed 1200 mm/s, layer thickness 30 µm, hatching distance 0.1 mm and stripe scanning strategy.	Tensile	Effect of build plate heating on the tensile properties was evaluated and build plate heating temperatures of 140 °C and 170 °C were reported to yield highest tensile strength of 408 ± 5 MPa and a yield strength of 284 ± 3 MPa. The platform heating induces artificial ageing within the print-parts during processing. Low platform heating temperatures result in under-ageing whereas high platform heating temperatures result in over-ageing of the printed samples.	EOS M270 Dual Mode system	[32]
Awd et al. (2018)	AlSi_10_Mg	Process parameters not disclosed. Build orientation: 0°, 45° and 90° with respect to the build plate.	Tensile	The pore density of 90° built samples were seven times higher than the 0° built samples. The 0° built samples displayed 8% higher tensile strength than the samples built at 90° orientation. Samples fabricated along the 45° orientation had a significantly reduced fracture strain.	M2 Cusing system	[33]
Baitimerov et al. (2018)	AlSi_12_	Laser power 200 W, layer thickness 50 µm, stripe hatch scanning strategy, powder dried at 100 °C for 1 h and the oxygen inside the build chamber was maintained below 500 ppm.	Density	Using the optimised process parameters, a relative density > 99% was achieved. The AlSi_12_ powder with good flowability and apparent density leads to good processability in SLM, while highly fine and spherically shaped powder particles result in poor processability. The particles with nearly spherical shaped morphology exhibited very poor flowability, leading to high porosity levels.	Sinterstation^®^ Pro DM125 SLM System	[34]
Bao et al. (2020)	AlSi_10_Mg	Laser power ranging 360 W–400 W, layer thickness 0.05 mm, laser scanning speed 1200 mm/s–1500 mm/s, scan spacing 0.13–0.16 mm.	Fatigue	The cyclic plasticity occurs when the longitudinal strain exceeds 0.9. The failure of the printed parts was attributed to the presence of large density of secondary micro-voids. The elongation of original defects along the loading direction is 10 times faster than that along the lateral side.	BLT S310	[35]
Bassoli et al. (2018)	A357	Laser power 950 W, scan speed 2000 mm/s, spot size 400 µm, hatch distance 0.2 mm, layer thickness 50 µm, nitrogen gas chamber atmosphere, preheating temperature 200 °C and skin core scan strategy.	Tensile, Fatigue	A 0.2% proof stress of 192 ± 17 MPa was found for as-built A357 samples built in the vertical orientation. A fatigue strength of 60 ± 5.3 MPa was obtained.	X Line 2000R (Concept Laser)	[36]
Beretta et al. (2020)	AlSi_10_Mg	Laser power 350 W, hatch distance 0.13 mm, scan speed 1650 mm/s, layer thickness 50 µm, preheating of build platform up to 150 °C. Build orientation: Horizontal (0°), Inclined (45°), and Vertical (90°).	Fatigue strength	Moderate correlation between the surface features at the origin of the fracture and surface roughness of the printed parts was found. Horizontal-built specimens had highest fatigue strength compared to inclined- and vertical-built specimens. EIFS and fatigue life prediction models were developed and verified using experimental data. The estimated cycles to failure was found within factor of 0.5–2 of the experimental values.	SLM 280HL v1.0	[37]
Biffi et al. (2018)	AlSi_10_Mg	Pulsed wave laser: laser power 300 W, build platform temperature 170 °C, exposure time 120 µs, point distance 130 µm, spot size 130 µm, hatch distance 140 µm, layer thickness 25 µm and build chamber environment argon. Continuous wave laser: laser power 350 W, build platform temperature 150 °C, laser travel speed 1150 mm/s, spot size 80 µm, hatch distance 170 µm, layer thickness 50 µm and build environment argon.	Compression	The continuous wave (CW) and pulsed wave (PW) showed similar Si network, however they varied in liquid melt pool. The continuous wave yielded better strength in compression, which resulted in reduced influence of the sample build orientation on compression behaviour. The samples built with CW laser emission was found to have better compressive strength than those processed with PW. The anisotropy is better exhibited in PW than CW.	Renishaw AM 250 system SLM Solutions SLM 500 system	[38]
Boschetto et al. (2017)	AlSi_10_Mg	Laser power 370 W, scan speed 1300 mm/s, hatch spacing 190 µm, layer thickness 30 µm, laser beam diameter 100 µm, hatch rotation 67°.	Roughness	Surface roughness prediction model that incorporates the staircase effect and defects arising due to satellites particles in the feedstock powder and the balling effect was developed. A case study on the surface roughness of a Pelton bucket was carried out, comparing the experimental values of surface roughness of curved complex surfaces to that of the model predicted data.	EOSINT^®^ M290	[39]
Brandl et al. (2012)	AlSi_10_Mg	Laser power 250 W, beam diameter 0.2 mm, layer thickness 50 µm, scanning speed 500 mm/s, scan spacing 0.15 mm and argon gas shielding, with and without build plate heating, build plate heating temperature 300 °C. Build orientation: 0°, 45°, and 90°.	Fatigue (using tensile samples), fracture analysis	Post heat treatment was found to have the most considerable effect, whereas the build orientation was found to have the least considerable effect on fatigue resistance. Heating the build plate to 300 °C tended to neutralize the anistotropy effects due to build orientation of the samples as well as enhanced the fatigue resistance. The fatigue limit and the static tensile strength significantly correlate with each other.	Trumpf TrumaForm LF130 powder-bed machine	[40]
Buchbinder et al. (2013)	AlSi_10_Mg	Laser power 195 W, layer thickness 30 µm, scanning speed 800 mm/s, hatch distance 150 µm, hatch rotation 67°, preheating temperatures 100°, 150°, 200°, and 250°.	Distortion	Distortions in part geometry reduced from 10.6 mm to nearly zero after preheating at 250 ℃. Reduction in hardness along with better resistance to crack growth was observed when the build plate was preheated. With preheating, lower temperature gradient results in lower thermal stresses in the printed parts.	EOSINT M 270	[41]
Casati et al. (2018)	AlSi_10_Mg	Laser power 340 W, hatch distance 0.2 mm, scan rate 1300 mm/s, layer thickness 30 µm, platform temperature 160 °C.	Tensile	Ageing of parts printed on cold-platform resulted in higher hardness and tensile strength compared to hot-platform printed and aged parts. The standard T6 temper heat treatment resulted in higher ductility but lower mechanical strength.	EOS M290 SLM system	[42]
Ch et al. (2019)	AlSi_10_Mg	Laser power 370 W, scan speed 1300 mm/s, hatch spacing 0.19 mm, layer thickness 30 µm, preheating up to 300 °C. Build orientation: Horizontal and Vertical.	Tensile, surface roughness, porosity, fracture morphology	The samples built under both argon and nitrogen exhibited similarity in defect structure, microstructure and nearly isotropic behaviour in mechanical properties. In the vertical orientation under nitrogen atmosphere, samples exhibited tensile strength of 385 ± 5 MPa, whereas, at horizontal orientation under argon atmosphere, samples exhibited the lowest strength of 338 ± 2 MPa. The meso-structure and microstructure of the material revealed the characteristic nature of SLMed AlSi_10_Mg alloy which contains melt pool tracks, columnar grains and cellular structure.	EOS M280 machine	[43]
Chen et al. (2017)	AlSi_10_Mg	Laser power 350 W, scan speed 1170 mm/s, layer thickness 50 µm, hatch spacing 240 µm.	Tensile	Orowan looping leads to high strength and high strain hardening capability. The hierarchical structure yields high yield strength of 300 MPa and UTS of 455 MPa.	SLM 250HL, SLM Solutions	[44]
Chou et al. (2015)	AlSi_12_	Laser power 0.5–4.5 kW, travel speed 90–180 mm/min, spot size 150 µm, hatch distance 0.1 mm, layer thickness 0.1 mm.	Hardness, density	Using pulsed laser for printing was demonstrated wherein Si refinement to size below 200 nm was achieved. The print part density was 95% and hardness was 135 HV when using pulsed laser for fabrication.	In house built Pulsed-SLM facility	[45]
Chou et al. (2018)	A356 (AlSi_7_Mg_0.3_)	Laser power (peak) 5 kW, square pulse, hatch distance 0.1 mm, layer thickness 100 µm, bidirectional zigzag scanning pattern, argon atmosphere.	Density	Dense parts (98 ± 0.4%) were fabricated using customized pulsed laser SLM system. Hunt’s criterion predicted primarily a columnar grain structure throughout the pulsed laser printed parts which was confirmed through electron backscatter diffraction (EBSD) observations.	In house built Pulsed-SLM facility	[21]
Dai and Gu (2016)	AlSi_10_Mg composite	Laser power 180 W, scan speed 100, 200, 300, and 400 mm/s, laser energy per unit length (LEPUL) 450, 600, 900, and 1800 J/m.	Numerical simulation, thermodynamics of the molten pool	The oxidation played a significant role in the melt pool dynamics with thermo-capillary convection where convection took place from inward to outward flow pattern. The thermocapillary flow yields a driving force for the migration and rearrangement of reinforcing particles in the melt pool.	In house SLM	[46]
Dai et al. (2018)	AlSi_10_Mg composite	Laser power 200 W, layer thickness 30 µm, scanning speed 100–300 mm/s, hatch spacing 60–100 µm.	Wear	The wear resistance was significantly influenced by the densification rate, distribution state and the powder particle size. Lowest wear rate of 3.4 × 10^−4^ mm^3^ N^−1^ m^−1^ was found when using an optimised laser volume energy density of 420 J/mm^3^.	In house SLM	[47]
de Menezes et al. (2019)	A357	Laser power 200 W, layer thickness 25 µm, hatch distance 115 µm. Isothermal aging at 160 °C for 4 h.	Tensile	The yield strength of the horizontal-built samples was considerably greater than that of the vertical-built samples. The yield strength of the samples in artificially aged condition was greater than the as-built condition. A crack orthogonal to the building orientation results in a significant decrease in fracture toughness of the printed samples.	Renishaw AM250 SLM system	[48]
Delahaye et al. (2019)	AlSi_10_Mg	Laser power 175 W, scan speed 195 mm/s, layer thickness 60 µm, preheating temperature 200 °C.	Nanohardness, fracture morphology, tensile	The fracture likely occurs in the heat-affected zone (HAZ) region due to the presence of coarse non-coherent Si precipitates. The intercellular network of eutectic structure is partially broken in the HAZ regions, thereby facilitating dislocation slip and plastic strain localisation. The cells size of the HAZ was found to match with the size of the dimples on the fracture surfaces.	MTT SLM 250	[49]
Deleroisse et al. (2017)	AlSi_10_Mg	Laser power 250 W, scan speed 571 mm/s, layer thickness 60 µm, and argon atmosphere. Heat treatment: T6 heat treatment, annealing at 525 °C for 5 h, water quench and then artificial ageing at 165 °C for 7 h.	Porosity, strut properties	Large levels of porosity and heterogenous microstructure was observed in inclined strut. Amount of hydrogen detected in the samples was larger than the hydrogen solubility in the liquid melt pool. Heat treatment suppresses microstructural heterogeneities.	-	[50]
Denti (2018)	A357	Laser power 950 W, scanning speed 2000 mm/s, hatch distance 0.2 mm, laser spot 400 µm, nitrogen chamber atmosphere, preheated at 200 °C and skin-core scan strategy.	Shear, Tensile	Yield strength (195 ± 17 MPa) and modulus (77.8 ± 6.8 GPa) were found to be maximum at 45° inclined samples. The ultimate tensile strength and shear strength of vertical-built samples were found to be the highest, 305 ± 15 MPa and 192 ± 9 MPa, respectively.	Concept Laser X Line 2000 R	[51]
Ding and Wang (2017)	AlSi_10_Mg	Laser power 400 W, laser beam radius 67.5 µm, hatch space 130 µm, scan speed 300 mm/s.	Modelling and simulation, density, hardness	There is negligible effect of exposure time on heat transfer, fluid flow, and melt pool dimension. The simulated temperature and melt pool dimensions increase with an increase in point distance.	-	[52]
Dong et al. (2018)	AlSi_10_Mg	Laser power 370 W, laser spot size 100 µm, hatch spacing 190 µm, scanning speed 1500 mm/s, layer thickness 30 µm, argon chamber atmosphere.	Tensile, porosity, fractography	The size effect has a significant influence on the geometric imperfection for SLM-processed AlSi_10_Mg strut. The porosity level of the sample decreased from 1.87% to 0.1% with the variation of build size from 1 mm to 5 mm. The overall strength and modulus decreased by approximately 30% with decrease in the build size.	Concept X-line 1000R machine	[53]
Everitt et al. (2016)	AlSi_10_Mg	Laser power 100 W, scan speed 250 mm/s, hatch space 50 µm, argon atmosphere.	Nanohardness	Chemical composition mapping and nano-indentation showed higher hardness in the SLM material compared to its cast counterpart. The printed samples exhibited a nanohardness of 9 ± 1 GPa, compared to 0.97 ± 1 GPa for the cast samples.	Realizer SLM-50	[54]
Fefelov et al. (2017)	AlSi_12_	-	Fracture surface morphology, tensile	The samples demonstrated yield and tensile strengths of about 102 MPa and 425 MPa, respectively, along with fracture strain of 12%.	EOSINT M 280 device	[55]
Fiegl et al. (2019)	AlSi_10_Mg	Laser power 1000 W, layer thickness of 50 µm, volume energy 40 J/mm^3^, island scanning strategy and nitrogen atmosphere.	Tensile	The impact of preheating on tensile strength in x/y plane seems to be more pronounced than the impact of beam deflection. UTS was found to be higher without preheating, 377 ± 16 MPa, compared to samples printed with preheating strategy, 355 ± 15 MPa.	X LINE 2000R (modified)	[56]
Fiocchi et al. (2017)	AlSi_10_Mg	Laser power 300 W, spot size 130 µm, layer thickness 25 µm, hatch distance 140 µm, and argon atmosphere.	Differential scanning calorimetry, hardness	Two exothermic phenomena involving Mg_2_Si precipitation and Si diffusion were found. Isothermal transformation temperatures were identified to be 263 °C and 294 °C.	Renishaw AM250	[57]
Fousova et al. (2018)	AlSi_10_Mg	Laser power 350 W, layer thickness 50 µm, scanning speed 500 mm/s, hatch spacing 170 µm, argon atmosphere and chessboard scanning strategy.	Tensile	Printed samples subjected to elevated temperatures underwent microstructural and mechanical property changes. At the tested temperature of 160 °C, a slight increase in tensile yield strength (by 5%) but also a strong reduction in elongation (by 60%) were observed.	SLM Solution 280HL machine	[58]
Gokuldoss Prashanth et al. (2016)	AlSi_12_	Laser power 320 W, scan speed 1455 mm/s, layer thickness 50 µm, hatch spacing 110 µm and argon atmosphere.	Tensile	The as-built samples exhibited a yield strength of 240 ± 1 MPa and UTS of 385 ± 4 MPa. After annealing at 300 °C, the yield strength and UTS reduced to 138 ± 3 MPa and 205 ± 5 MPa, respectively. The softening behaviour of the Al matrix was inversely proportional to strain hardening behaviour.	SLM 250 HL	[18]
Girelli et al. (2019)	AlSi_10_Mg	Laser power 400 W, beam diameter 100 µm, argon atmosphere. Build orientation: Horizontal and Vertical Heat treatment: Solutionising at 540 °C for 1–9 h, followed by ageing at 160 °C for 4 h or 180 °C for 2 h.	Impact properties, fracture morphology	The fracture in the horizontal samples propagated along the hatch overlap regions whereas in the vertical samples the crack propagation was primarily along the inter-layer regions. The building orientation has a significant influence on the properties of the printed samples even after post heat treatment is applied. After hot isostatic pressing the samples, the porosity reduced considerably, and the fracture toughness of the specimens improved.	EOS M290 system	[59]
Gong et al. (2017)	AlSi_10_Mg	-	Tensile	The mechanical properties of AlSi_10_Mg parts produced by SLM are much higher than the alloy formed by conventional casting.	-	[60]
Hadadzadeh et al. (2018)	AlSi_10_Mg	Laser power 370 W, beam spot size 100 µm, scanning speed 1300 mm/s, hatch spacing 190 µm, layer thickness 30 µm, argon atmosphere. Build orientation: Horizontal and Vertical.	Dynamic compression	Columnar cellular structure was observed in the horizontal-built samples, whereas equiaxed structure was observed in the vertical-built samples. The dislocation density in the as-built horizontal and vertical samples was 1.14 × 10^14^ m^−2^ and 3.05 × 10^14^ m^−2^, respectively. These dislocation networks converted to low angle grain boundaries through dynamic recovery process to reduce the energy, thereby resulting in softening of the samples.	EOS M290 machine	[61]
Hadadzadeh et al. (2019)	AlSi_10_Mg	Laser power 370 W, scanning speed 1300 mm/s, hatch spacing 190 µm, layer thickness 30 µm, stripe scanning strategy, argon atmosphere.	Dynamic compression	The samples exhibited strain hardening at strain rates lower than 1400 s^−1^, and at higher strain rates they displayed double peak flow stress deformation regime. Strengthening mechanisms included Hall-Petch contributed by cell walls, Orowon contributed by Si precipitates, and dislocation hardening. Both cell walls and Si precipitates contribute to impeding the dislocation motion and development of dislocation networks.	EOS M290	[62]
Han and Jiao (2019)	AlSi_10_Mg	Laser power 200 W, hatch spacing 130 µm, layer thickness 25 µm, scanning speed 500 mm/s, chessboard bidirectional scanning strategy. Heat treatment: Solution-treated at 550 °C for 2 h, followed by furnace cooling. Laser surface remelting treatment: Laser power 20 W, beam diameter 32 µm, scanning speed 300 mm/s, layer thickness 0.01 mm.	Tensile, Hardness, Roughness	The surface roughness of the laser surface remelted samples was significantly reduced (0.93 µm) compared to the as-built samples (19.3 µm). The yield strength reduced from 200 MPa in the as-built samples to 100 MPa in the heat-treated samples. The laser surface remelting process improved the microhardness by 19.5% through refining the microstructure.	Renishaw AM250	[63]
Hatamleh et al. (2018)	A357	Heat treated at 170 °C for 3 h. Laser shock peening applied.	Finite element analysis, residual stress analysis, tensile testing	Laser shock peening induced tensile residual stresses within certain regions of the samples. Tensile tests were conducted, and the results were evaluated with finite element analysis by using Johnson-Cook parameters.	-	[64]
Hitzler et al. (2016)	AlSi_10_Mg		Tensile	Post heat-treatment resulted in homogeneous microstructure, thereby exhibiting remarkable improvement in ductility.	SLM 280HL machine	[65]
Hitzler et al. (2017)	AlSi_10_Mg	Laser power 350 W, hatch spacing 0.42 mm, scan speeds 600–930 mm/s, base plate heating 200 °C. Build orientation: horizontal, inclined (45°), and vertical.	Fracture toughness	The fracture toughness was found to be 40.63 KMPa.m^0.5^. The vertical-built samples were found to have poor fracture toughness.	SLM 280HL machine	[66]
Hitzler et al. (2017)	AlSi_10_Mg	Laser power 350 W, hatch spacing 0.42 mm, scan speeds 600–930 mm/s, base plate heating 200 °C, nitrogen atmosphere.	Tensile, surface hardness	The elastic modulus, tensile strength, and elongation were found to be sensitive to the part print orientation. For the non-heat-treated condition, the Young’s modulus varied from 62.5 to 72.9 GPa, the Poisson’s ratio ranged from 0.29 to 0.36, UTS from 314 MPa to 399 MPa, and elongation from 3.2% to 6.5%. The heat-treated samples exhibited dissimilar responses to an additional aging procedure and were dependent on the build height.	SLM 280HL machine	[67]
Hitzler et al. (2018)	AlSi_10_Mg	Laser power 400 W, layer thickness 50 µm, nitrogen atmosphere.	Tensile, compression	Young’s modulus of the printed samples exceeded the nominal value of ∼70 GPa. Vertical-built samples showed the least elastic modulus. A maximum compressive strength value of 530 MPa was attained.	SLM 280HL machine	[68]
Hu et al. (2016)	AlSi_10_Mg	Laser energy input 700–1300 J/m, layer thickness 50 µm, argon atmosphere.	CFD model, thermal gradient	The results show that the temperature and molten pool depth gradually increases with new layers and energy input. The cooling rate increases progressively while the solidification morphology parameter decreases with the increase of energy input. The increase in energy input lead to non-linear increase in melt pool depth and remelting depth.	-	[69]
Iturrioz et al. (2018)	AlSi_10_Mg	Laser power 400 W, protective atmosphere. Heat treatment: Stress-relieving (heating rate of 10 °C/min until 300 °C for 2 h), Solution-treated at 450 °C and 550 °C for 2 h, followed by water quenching, and T6 heat treatment.	Tensile, hardness, density	The as-built samples achieved a relative density of 99.9%. After heat treatments, Si forms polygonal precipitates whose size increases as the temperature increases. This results in a decrease in tensile strength and hardness with heating up to 450 °C. T6 heat treatment (550 °C for 2 h followed by ageing at 180 °C for 12 h) was found to be best performing, which yielded the tensile strength of 307 ± 8 MPa, hardness 101 ± 4 HV and ductility 9 ± 3%.	SLM 280HL machine	[70]
Jawade et al. (2020)	AlSi_10_Mg	Laser power 250 W, beam diameter 0.2 mm, layer thickness 50 µm, scanning speed 500 mm/s.	Tensile	The horizontal build orientation yielded better tensile strength when compared to samples built in the vertical orientation. Furthermore, microstructure revealed anisotropy and its dependency on build orientation. The UTS was found to be maximum on horizontal orientation (401.89 MPa), yield strength (301.26 MPa) and elongation of 4.3%, which is superior than the counterpart at vertical orientation.	Renishaw AM 250	[71]
Jiang et al. (2018)	AlSi_10_Mg composite	Laser power 370 W, scan speed ranging from 900–1900 mm/s, scan spacing 105 µm, layer thickness 30 µm, zigzag scan strategy.	Tensile, hardness, density	The hardness and tensile strength improved by approx. 10% and 20%, respectively, with the addition of 1 wt% CNTs.	Concept Laser M2 Cusing SLM system	[72]
Jing et al. (2019)	AlSi_10_Mg	Hatch spacing 190 µm, layer thickness 30 µm, volume energy density 26–54 Jmm^−3^, laser power 310–390 W, scanning speed 1300–1900 mm/s, chamber pressure 0.45 mbar. Build orientation: Horizontal and Vertical.	Porosity, microhardness, tensile testing and fracture morphology	Microstructure of the samples printed using optimised process parameters consisted of extremely fine Al-Si eutectic dispersed within primary α-Al dendrites. Hardness in the transverse direction was higher than that along the longitudinal direction. The engineering stress-strain curve at different orientations also shows anisotropy in the strength and ductility, which is possibly due to the orientation between the tensile stress and crystal growth direction.	EOS 83 M290 SLM	[73]
Kang et al. (2017)	AlSi_12_	Laser power 400 W, scanning speed 5000 mm/s, layer thickness 50 µm and hatch distance 45 µm.	Tensile	Ultrafine microstructure consisting of cellular α-Al and nano-sized Si particles was found due to the high cooling rate of the SLM process As the laser scanning speed increases, both tensile strength and ductility of SLM processed samples decrease. Higher energy density was required to fabricate dense samples from in-situ SLM fabrication using elemental Al and Si powders, compared to pre-alloyed Al-12Si powders.	Realizer SLM 250 machine	[74]
Kempen et al. (2012)	AlSi_10_Mg	200 W fiber laser, laser beam diameter 150 µm, scan speed 1400 mm/s, scan spacing 105 µm. Build orientation: Horizontal and Vertical.	Tensile	Mechanical properties of printed parts comparable or even exceeding those of conventionally cast AlSi_10_Mg. Maximum UTS observed for vertical samples 396 ± 8 MPa, but the elongation was better for the horizontal orientation 5.55 ± 0.4%.	Modified Concept Laser M1 SLM Machine	[4]
Kempen et al. (2014)	AlSi_10_Mg	Laser power 300 W, laser spot size 80 µm, scan speed 1600 mm/s. Build orientation: XY and Z.	Surface roughness, Metrology	Samples printed in the Z-orientation has to be compensated for the width of the melt pool. Samples printed in the XY-orientation showed dross formation.	In-house built SLM machine	[75]
Kempen et al. (2015)	AlSi_10_Mg	200 W fiber laser, laser beam diameter 150 µm, scan speed 1400 mm/s, scan spacing 105 µm and layer thickness 30 µm. Build orientation: Horizontal and Vertical.	Tensile, Hardness	Mechanical properties of printed parts comparable or even exceeding those of high pressure die cast AlSi_10_Mg. Hardness found to be 152 ± 5 HV after ageing at 175 °C for 6 h.	Modified Concept Laser M1 SLM Machine	[76]
Kempen et al. (2011)	AlSi_10_Mg	170 to 200 W fiber laser, laser beam diameter 150 µm, scan speed 200–1400 mm/s, scan spacing 105 µm and layer thickness 30 µm. Build orientation: Horizontal and Vertical.	Density, Roughness	Narrow process window to obtain optimal density and surface quality for printed parts. Higher scan speed of 1400 mm/s was used for high density/productivity demands, whereas lower scan speeds of 1100–1200 mm/s were used for parts with a high demand in top surface quality.	Modified Concept Laser M1 SLM Machine	[77]
Kim et al. (2016)	AlSi_10_Mg	Laser power 200 W, scanning speed 1500 mm/s, bidirectional scanning vector.	Tensile	Significant stress partitioning exists between the Si (680 MPa) and Al (260 MPa) constituents. The fracture analysis reveals large voids and cracks in the microstructure, particularly in the soft Al matrix under loading.	Concept Laser M2 machine	[78]
Kimura and Nakamoto (2016)	A356	Laser power 200–370 W, scan speed 400–3000 mm/s, scan spacing 0.08–0.18 mm, layer thickness of 30 µm.	Tensile, Density	Relative density of 99.8% could be obtained by optimizing the laser irradiation conditions. Tensile strength of 400 MPa, yield strength of 200 MPa, and elongation of 12–17% was obtained for optimally printed samples. After annealing the UTS reduced to 250 MPa and yield strength to 125 MPa, however, elongation increased to 30%.	EOSINT M280 SLM	[20]
Krishnan et al. (2014)	AlSi_10_Mg	200 W Yb fiber laser, layer thickness 20–40 µm, beam diameter 0.1 mm, laser power 180–195 W, scanning speed 700–900 mm/s, hatch distance 0.1–0.25 mm, nitrogen atmosphere.	Hardness, Density	Hatch distance is the most influential process parameter affecting the print part density. There is a close correlation between the geometry of scan tracks and macroscopic properties of the printed parts. The optimal energy density was found to be 1.2–1.8 J/mm^2^.	-	[79]
Li et al. (2016)	AlSi_10_Mg	Laser power 350 W, beam diameter 80 µm, scan speed 1140 mm/s, layer thickness 50 µm, scan spacing 170 µm, argon chamber atmosphere, substrate plate pre-heated to 100 °C. Heat treatment: T6 treatment, solution treated at 450 °C, 500 °C, and 550 °C for 2 h, water quenched, and aged at 180 °C for 12 h.	Tensile, hardness	The solubility of Si in the Al matrix of the as-built samples was found to be about 8.89 at%, which significantly decreased after solution heat treatment, and further decreased with ageing treatment. The tensile strength decreased from 434.25 ± 10.7 MPa for the as-built specimens to 168.11 ± 2.4 MPa for the specimens that were solution heat-treated only. However, the fracture strain increased from 5.3 ± 0.22% to 23.7 ± 0.84%.	SLM250 HL	[80]
Li et al. (2017)	AlSi_10_Mg	Build orientation: Horizontal (Y-direction) and Vertical (Z-direction), tensile testing carried out at −71 °C.	Tensile	The tensile strength and elongation of the horizontally-built samples are 340 MPa and 11.2%, respectively, compared to 350 MPa and 13.4%, respectively, for the vertically-built samples. The microstructure revealed fish-scale morphology along the Y-build direction and oval shaped structure in the Z-build direction.	-	[81]
Li et al. (2015)	AlSi_12_	Laser power 200 W, scanning speed 500 mm/s, layer thickness 50 µm, scan spacing 150 µm, substrate heated to 200 °C, argon atmosphere. Heat treatment: Solution treated at 500 °C for 4 h and water quenched.	Tensile, residual stresses, FEM	Spherical Si particles with size less than 100 nm formed at the Al grain boundaries. However, the coarse and fine Si precipitates were found to be homogeneously distributed in the Al matrix. The ultrafine eutectic microstructure yields significantly better tensile properties and an extremely high ductility of approx. 25% after solution heat treatment.	ReaLizer SLM-100 machine	[82]
Li et al. (2018)	AlSi_10_Mg	Laser power 200 W, hatch spacing 80 µm, laser spot diameter 80 µm, layer thickness 25 µm, exposure time 100–180 µs, point distance 60–100 µm.	Modelling and simulation	A 3D FE model was developed to simulate the thermal behaviour and melt pool dimensions of the printed parts. The cooling rate of the melt pool reduced from 7.93 × 10^6^ °C/s to 3.61 × 10^6^ °C/s as the laser exposure time is increased from 100 µs to 180 µs. Alternatively, with an increase in point distance from 60 µm to 100 µm, the cooling rate increased from 3.25 × 10^6^ °C/s to 7.48 × 10^6^ °C/s. The melt pool size (width and depth) increased as the laser exposure time is increased, and vice-versa when the point distance is increased.	Renishaw AM 400 SLM system (Renishaw PLC)	[83]
Li and Gu (2014)	AlSi_10_Mg	Laser power of 150–300 W, laser spot diameter 70 µm, scanning speed 100–400 mm/s, layer thickness 50 µm, hatch spacing 50 µm, argon atmosphere.	Modelling and simulation	SLM experimental results validated the obtained thermal behavior in simulation. The cooling rate increased from 2.13 × 10^6^ °C/s to 2.97 × 10^6^ °C/s when the laser power was increased from 150 W to 300 W, and from 1.25 × 10^6^ °C/s to 6.17 × 10^6^ °C/s when the scan speed increased from 100 mm/s to 400 mm/s. A sound metallurgical bonding between the neighboring layers was obtained at the optimized combination of process parameters, including laser power of 250 W and scan speed of 200 mm/s.	YLR-500-SM	[84]
Liu et al. (2010)	AlSi_10_Mg	Laser power 200 W	Tensile, hardness, density, roughness	An 80% decrease in surface roughness was observed when the printed parts were sand blasted. The tensile properties of the as-built samples tested at both room temperature and high temperature (200 °C) are significantly better than the conventionally heat-treated cast parts.	M2 Cusing from Concept Laser, Germany	[85]
Liu et al. (2018)	AlSi_10_Mg	Laser power 500 W, scan speed 1500 mm/s, layer thickness 30 µm, zigzag scan strategy, scanning angle alternated by 90° for subsequent layers.	Hardness, wear resistance, modelling and simulation	The cooling rate for the layers printed at the top region of the sample was determined to be about 1.44 × 10^6^ K/s, which is significantly higher than that experienced by the bottom layers (≤ 1 × 10^3^ K/s). The top surface area has a lower degree of crystallinity of Al matrix than that of core area. The surfaces of the printed samples exhibited higher hardness and wear resistance compared to the core regions.	BLT-S300 SLM machine	[86]
Liu et al. (2018)	AlSi_10_Mg	Laser power 160–200 W, spot size 100 µm, scanning speed 200–1000 mm/s, argon atmosphere.	Modelling and simulation	The cooling rate, temperature gradient and the solidification rate increase with an increase in laser power and decrease with an increase in scanning speed. Along Z direction, the cooling rates and the temperature gradient are lower compared to Y direction.	IPG YLR-200	[87]
Louvis et al. (2011)	AA 6061	Laser power 50–100 W, laser spot size 80 µm, hatch spacing 0.05–0.3 mm, scan speed range 75–1000 mm/s, layer thickness 50 µm, argon atmosphere, point distance 65–85 µm	Density	The formation of oxides can be avoided by using high laser powers while printing. The maximum relative density was obtained at high laser power of 100 W (89.5%).	MCP Realizer 100 SLM machine	[8]
Maamoun et al. (2019)	AlSi_10_Mg & AA 6061	For AA 6061: Laser power 300–370 W), scanning speed 800–1300 mm/s, hatch spacing 0.1–0.19 mm, energy density 40.5–123.3 J/mm^3^. For AlSi_10_Mg: Laser power 200–370 W), scanning speed 1000–1300 mm/s, hatch spacing 0.15–0.25 mm, energy density 27–65 J/mm^3^.	Tensile, hardness	The microstructure of AA 6061 parts did not show the same fibrous Si network that formed inside the AlSi_10_Mg microstructure due to lower Si content. The size of the melt pool increases with an increase in energy density. An energy density range of 50 to 60 J/mm^3^ was found to be optimum to significantly minimize the formation of keyhole defects and porosities. Maximum tensile strength of both the AlSi_10_Mg and AA 6061 printed samples was 396.5 MPa.	-	[88]
Maconachie et al. (2020)	AlSi_10_Mg	Laser power 350 W, scanning speed 1650 mm/s, layer thickness 30 µm, hatch spacing 130 µm and beam diameter 90 µm. Build Orientation: 0°, 45° and 90°.	Tensile, dynamic behaviour in tension (SHTB), fracture	The tensile strength is not significantly affected by the build orientation of the printed samples. Samples built perpendicular to the build direction failed at greater strain than those built parallel to the build direction. The anisotropic properties of the samples were insensitive to the strain rate applied during mechanical testing.	SLM500	[89]
Majeed et al. (2019)	AlSi_10_Mg	Laser power 320 W, hatch spacing 0.08 mm, layer thickness 30 µm, scanning speed 900 mm/s, checkerboard scanning strategy, argon atmosphere. Heat Treatment: T4 treatment involving solution heat treated at 530 or 540 °C for 2 h and water quenched. T6 treatment involving solution treated at 530 °C for 2 h, water quenched and artificially aged at 155 °C for 12 h.	Density, tensile, fracture	As the wall thickness increased from 0.5 mm to 1.5 mm, an increase in porosity was observed. Ageing heat treatment resulted in better density of the thin wall samples. The size of the pores increased till the wall thickness of 1.5 mm and then decreased with further increase in wall thickness.	280 HL SLM	[90]
Maskery et al. (2015)	AlSi_10_Mg	Laser power 200 W, layer thickness 25 µm, hatch spacing 130 µm, scanning speed 571 mm/s, point distance 80 µm, exposure time 140 µs, build platform temperature 180 °C, hatch rotation 67°. Heat treatment: Solution treated at 520 °C for 1 h, water quenched, and aged at 160 °C for 6 h.	Tensile, fatigue	The elongation at break for the heat-treated material was nearly three times greater than that observed for the as-built material, and the fatigue strength at 10^6^ cycles was around 1.6 times as high. The UTS was reduced from 330 ± 10 MPa to 292 ± 4 MPa and the ductility enhanced from (1.4 ± 0.3)% to (3.9 ± 0.5)%.	Renishaw AM250 SLM machine	[91]
Mfusi et al. (2019)	AlSi_10_Mg	Laser power 150 W, scanning speed 1000 mm/s, hatch spacing 50 µm, and layer thickness 50 µm. Build Orientation: XY, 45°, and Z orientations. Heat treatment: Stress relief treatment at 300 °C for 2 h followed by furnace cooling.	Tensile, fatigue crack growth, fracture toughness, density, hardness, porosity	Anisotropy due to different build orientations was found even after post stress relief heat treatment. Maximum relative density was reported for XY printed samples (97.33 ± 0.92%). However, maximum hardness was reported for the 45° orientation printed samples (47.32 ± 3.35 HV).	SLM Solution M280 GmbH	[92]
Ngnekou et al. (2017)	AlSi_10_Mg	Laser powers 200 W and 400 W. Build Orientation: XY and Z Heat treatment: Stress relieved at 160 °C for 1 h or 300 °C for 2 h, T6 heat treatment involving solution treated at 540 °C for 8 h, water quenched, and tempered at either 20 °C for 24 h or 160 °C for 10 h.	Fatigue	The improvement in fatigue resistance is less pronounced when large-sized defects are present in the printed samples. There is no influence of the defect type on the fatigue limit.	Phenix PM100 3D Systems machine EOS M290 machine	[93]
Nurel et al. (2018)	AlSi_10_Mg	Laser power 400 W, spot diameter 80 µm, scan velocity 1000 mm/s, strip scanning strategy, hatch distance 200 µm, hatch rotation 67°, layer thickness 60 µm, argon atmosphere, build plate temperature 35 °C. Build orientation: Horizontal and Vertical Heat Treatment: T5/Stress relief treated at 300 °C for 2 h.	Dynamic-Compression	The dynamic anisotropic properties were insensitive to variation in strain rates. Anisotropic differences were considerably reduced by applying T5 heat treatment. The as-built samples failed after SHPB tests which was observed in the T5 heat treated samples.	EOSINT M280 system	[94]
Nurel et al. (2018)	AlSi_10_Mg	Laser power 400 W, spot diameter 80 µm, layer thickness 30 µm. Build orientation: Horizontal and Vertical. Heat treatment: T5 at 300 °C for 2 h.	Dynamic-Compression	No strain sensitivity was observed. True stress for as-built and heat-treated conditions are 569 ± 8.5 MPa and 427 ± 4.8 MPa respectively.	EOSINT M280 system	[95]
Ojha et al. (2018)	AlSi_10_Mg	Laser power 100 W, scanning speed 2000 mm/s, spot size 0.2 mm, layer thickness 80 µm.	Modelling and simulation	FEA was carried out to investigate temperature evolution, heat transfer and solidification process. Simulation results are dependent on process parameters along with material properties.	-	[96]
Pei et al. (2017)	AlSi_10_Mg	Laser power 180 W, scanning speed 600–1600 mm/s, layer thickness 35 µm, spot size 70 µm.	Modelling and simulation	The perturbation or the instability within the molten pool results in the formation of pores during SLM, which have a direct influence on the densification level. At high scanning speed, the track morphology became discontinuous leading to poor bonding and balling.	SLM-150 equipment	[97]
Pola et al. (2019)	AlSi_10_Mg	-	Fatigue, porosity, surface roughness, tensile	Post-treatment is effective in reducing surface roughness and inducing compressive residual stresses on the material surface. Sand blasting had a beneficial effect of the fatigue resistance.	EOS M290 system	[98]
Ponnusamy et al. (2018)	AlSi_12_	Laser power 285 W, scanning speed 1000 mm/s, hatch spacing 100 µm, layer thickness 40 µm, defocus distance −4 mm, alternative scan strategy. Build orientation: Horizontal, Inclined, Vertical	Dynamic - Compression	The dynamic compressive strength increased with an increase in the angle of print orientation, i.e., from 0° to 90°. The yield strength and compressive strength decrease for printed samples tested at elevated temperature of 200 °C. The flow stress was found to be higher for dynamic loading compared to quasi-static loading at elevated temperature.	3D Systems ProX 200	[99]
Ponnusamy et al. (2020)	AlSi_12_	Laser power 285 W, scanning speed 1000 mm/s, hatch spacing 100 µm, layer thickness 40 µm, defocus distance −4 mm, alternative scan strategy, hexagon scan pattern, argon atmosphere. Build orientation: Vertical Heat treatment: Annealed at 200 °C and 400 °C for 3 hrs.	Dynamic compression	Thermal softening was observed in printed samples tested at elevated temperatures, which resulted in significant reduction in flow stress. A 20% and 50% reduction in flow stress was observed when samples were tested at 200 °C and 400 °C test temperatures, respectively. A 12% and 45% reduction in flow stress was observed for samples heat treated at 200 °C and 400 °C, respectively, and then tested.	ProX 200	[100]
Prashanth et al. (2014)	AlSi_12_	Laser power 320 W, layer thickness 50 µm, hatch spacing 110 µm, hatch rotation 73°, scanning speed 1455–1939 mm/s, argon atmosphere. Build orientation: 30°, 45°, 60°, 75°, 90°. Heat treatment: Solution treated at 473–723 K for 6 h.	Wear rate, corrosion properties	As-built samples exhibit better wear resistance and similar corrosion resistance compared to cast counterparts. Both wear and corrosion properties deteriorated with annealing post heat treatment, due to growth of Si precipitates.	SLM 250 HL	[16]
Prashanth et al. (2017)	AlSi_12_	Laser power 320 W, layer thickness 50 µm, hatch spacing 110 µm, hatch rotation 73°, scanning speed 1455–1939 mm/s, argon atmosphere, scan strategies included single melt, double melt, single melt continuous, and checkerboard, base plate heating 473–673 K. Heat treatment: Solution treated at 473–723 K for 6 h.	Tensile	The difference in tensile properties were attributed to the variation in crack propagation path. The samples printed without contour exhibited significant increase in ductility without compromising on the tensile strength. The results indicate that the room temperature tensile properties can be tuned (between YS: 115–290 MPa, UTS: 220–460 MPa and ductility: 2.8–9.5%) in-situ with appropriate selection of process parameters.	SLM 250 HL	[101]
Prashanth et al. (2014)	AlSi_12_	Laser power 320 W, layer thickness 50 µm, hatch spacing 110 µm, hatch rotation 73°, scanning speed 1455–1939 mm/s, argon atmosphere. Build orientation: 30°, 45°, 60°, 75°, 90°. Heat treatment: Solution treated at 473–723 K for 6 h.	Tensile	The Al and Si phases show remarkably small crystallite sizes of about 118 and 8 nm. The as-built samples exhibited a yield strength of 260 MPa and tensile strength of 380 MPa, which was significantly higher than the cast counterparts. The texture of the microstructure of the printed samples varied with variation in build orientations, however, this did not affect the tensile properties.	SLM 250 HL	[17]
Qiu et al. (2015)	AlSi_10_Mg	Laser power 150–400 W, beam spot size 50 µm, scanning speed 1000–7000 mm/s.	Compression	The dimensions of the melt pool increased with an increase in laser power resulting in strut diameters deviating from the designed values. The compressive load bearing capacity of the lattice structures increased with an increase in strut diameter. Deformation of lattice structures occurred by homogeneous deformation until the maximum stress was achieved after which the structure lost structural integrity via a series of shear banding events at around 45° to the compression axis.	Concept Laser M2 Cusing SLM system	[102]
Rakesh et al. (2018)	AlSi_10_Mg	Laser beam diameter 80 µm, argon/nitrogen gas inert atmosphere. Build orientation: Transverse (XZ), Longitudinal (Y)	Tensile, Impact strength	Printed samples displayed superior tensile strength (~350 MPa) and impact strength compared to cast parts. Significant improvements in tensile properties and impact energies were observed in the transversely-built samples irrespective of the chamber atmosphere.	EOSINT M280 machine	[103]
Rao et al. (2016)	A357	Laser power 100–370 W, layer thickness 30 µm, hatch distance 0.1 mm, spot size diameter 0.1 mm, scan speed 500–5000 mm/s, substrate temperature 35–200 °C. Build orientation: Horizontal, Vertical.	Tensile	The mechanical property was largely affected by different substrate temperatures. The coarse Si precipitates formed along the build direction facilitates intercellular failure, resulting in poor tensile properties.	EOSINT M280	[25]
Rao et al. (2017)	A357	Laser power 100–370 W, layer thickness 30 µm, hatch distance 0.1 mm, spot size diameter 0.1 mm, scan speed 500–5000 mm/s, substrate temperature 35–200 °C. Heat treatment: Stress relieving at 300 ± 1 °C and air-cooled, solution treating at 535 ± 3 °C in salt bath from 0.25 h to 150 h followed by water quenching.	Tensile	The as-built samples had an ultrafine microstructure, with high residual stresses and non-equilibrium solid solute concentration of Si in the supersaturated Al matrix. The tensile properties of the printed A357 samples were comparable or better than the traditional cast counterparts. The UTS and YS of as-built sample are 426.4 ± 2.6 MPa and 279.6 ± 1, respectively, however, the ductility was found to improve after stress-relieving (13.6 ± 0.6%).	EOSINT M280	[24]
Rashid et al. (2018)	AlSi_12_	Laser power 285 W, scan speed 1000–2000 mm/s, layer thickness 40 µm, hatch distance 100 µm, defocus distance –4 mm. Build orientation: Horizontal, Inclined (45°), Vertical	Tensile, density	Energy per layer in the range of 504–895 J yielded ≥ 99.8% relatively dense AlSi_12_ SLM-printed samples. Yield strength range 225–263 MPa, tensile strength range 260–365 MPa, and ductility range 1–4% was found for the printed samples with different build orientations. Anisotropy in mechanical properties was attributed to differences in relative densities.	ProX 200	[104]
Rashid et al. (2017)	AlSi_12_	Laser power 285 W, scan speed 1000–2000 mm/s, layer thickness 40 µm, hatch distance 100 µm, defocus distance –4 mm. Lattice structures: Circular cells, honeycomb cells, triangular cells	Flexural	The printed samples exhibited brittle failure. Triangular lattice structure had the highest flexural strength of 175.80 ± 1 MPa, circular 151.35 ± 0.67 MPa, and honeycomb 143.16 ± 3.85 MPa, whereas the solid specimen had a strength of 290 ± 26 MPa. Triangular lattice structure showed good flexural modulus of 5 GPa compared to the honeycomb structure (4.34 GPa) and circular structure (4.37 GPa).	ProX 200	[105]
Rathod et al. (2019)	AlSi_12_	Laser power 320 W, scan speed 1455 mm/s, layer thickness 50 µm, and hatch spacing 110 µm.	Wear	Printed samples displayed significant anisotropy in wear rate due to change in laser track orientation. Porosity significantly affected the wear rate.	280HL equipment	[106]
Raus et al. (2017)	AlSi_10_Mg	Laser spot size 80 µm, laser power 350 W, scan speed 1650 mm/s, layer thickness 30 µm, hatch distance 0.13 mm, stripe scanning strategy, argon atmosphere, build platform temperature 150 °C.	Tensile, hardness, impact toughness	Porosities of the order of 5 to 20 µm were observed in relatively dense (99.13%) sample. Printed samples displayed higher mechanical properties compared to high pressure die cast samples.	SLM 125 HL	[107]
Read et al. (2015)	AlSi_10_Mg	Laser power 100–200 W, laser track width 150 µm, layer thickness 30 µm, scan speed 700–2000 mm/s, hatch spacing 0.2–0.8 mm, island scanning strategy, island size 2–8 mm, argon atmosphere. Build orientation: Horizontal, Vertical.	Tensile, creep resistance	A critical energy density of 60 J/mm^3^ was found wherein minimum pore fraction was observed. Creep results showed better rupture life than cast alloy, displaying good agreement with the Larson–Miller literature data. Unmelted powder particles give rise to local cracking, as observed on the fracture surfaces.	Concept Laser M2 Cusing SLM system	[108]
Rosenthal et al. (2015)	AlSi_10_Mg	Laser power 400 W, laser spot size 100 µm, scan velocity 1000 mm/s, strip scanning strategy, hatch rotation 67°.	Tensile, density	The mechanical properties of the printed parts displayed a strong dependency on the microstructure and are comparable or higher than cast part after T6 heat treatment.	EOSINT M280	[109]
Rosenthal et al. (2018)	AlSi_10_Mg	Laser power 400 W, laser spot size 100 µm, scan velocity 1000 mm/s, hatch distance 200 µm, layer thickness 60 µm, argon atmosphere. Build orientation: Horizontal (X), Vertical (Z). Heat Treatment: T5 stress relief treatment at 300 °C for 2 h, modified T5 at 200 °C for 2 h.	Impact resistance	Horizontally built specimens absorbed more impact energy compared to vertically built specimens.	EOSINT M280	[110]
Rosenthal and Stern (2016)	AlSi_10_Mg	Laser power 400 W, laser spot size 100 µm, scan velocity 1000 mm/s, hatch distance 200 µm, layer thickness 60 µm. Heat treatment: Treated at 100–250 °C for 2 h, treated at 200 °C for 168 h, treated at 100 °C for 336 h.	Hardness	Results revealed that the heat treatments conducted in the range of 100 °C–300 °C displayed noticeable increase in hardness values due to precipitation/coarsening of the Si phase.	EOSINT M280	[111]
Rosenthal et al. (2014)	AlSi_10_Mg	Laser power 400 W, scanning speed 1000 mm/s, strip scanning strategy Build orientation: Vertical, Horizontal. Heat treatment: Stress relieving at 300 °C for 2 h.	Tensile, hardness, fracture morphology	The printed parts displayed room temperature mechanical properties comparable or even exceeding conventionally cast AlSi_10_Mg samples. In the vertical orientation, the samples display Young’s modulus of 69.5 to 73 GPa, yield strength 167–170 MPa, UTS 269–277 MPa and elongation ranging 7.8–8.7%, whereas in the horizontal orientation Young’s Modulus ranges between 69–71.3 GPa, yield strength 168–170 MPa, UTS of 267 MPa and elongation ranging 8.6–9.5%.	-	[112]
Rosenthal et al. (2017)	AlSi_10_Mg	Laser power 400 W, laser spot size 100 µm, scan velocity 1000 mm/s, hatch distance 200 µm, layer thickness 60 µm, argon atmosphere, build plate temperature 35 °C, strip scanning strategy, hatch rotation 67°. Build orientation: Vertical, Horizontal.	Tensile, fracture surface analysis	Printed samples were sensitive to strain rate variations with significant changes to the flow stress and strain hardening exponents with an increase in strain rate. The strain rate sensitivity was similar in both vertical and horizontally printed samples, while the true strain was significantly higher in the samples built in the horizontal orientation.	EOSINT M280	[113]
Samantaray et al. (2018)	AlSi_10_Mg	Laser power ranging 70–190 W, laser spot diameter 0.2 mm, scanning speed 100–500 mm/s, layer thickness 1 mm.	Modelling and simulation	The maximum temperature of the molten pool increased from 731 °C to 2672 °C and the molten pool length changed from 0.286 mm to 2.167 mm, when the laser power increased from 70 W to 190 W. The sintering depth of the powder layer increased with an increase in laser power but decreased when the scan speed was increased.	-	[114]
Siddique et al. (2017)	AlSi_12_	Laser power 400 W, volume energy density 39.6 J/mm^3^, argon atmosphere. Heat treatment: Stress relieving at 240 °C for 2 h followed by oven cooling.	Fatigue, tensile	The microstructure of the printed samples consisted of fine grains and precipitates that resulted in increased quasi-static strength compared to that of the cast counterparts. The fatigue strength of the as-built hybrid samples was comparably better than the as-built samples.	SLM 250	[115]
Siddique et al. (2015)	AlSi_12_	Laser power 350 W, scan speed 930 mm/s, hatch distance 0.19–0.25 mm, energy density 20–39.6 J/mm^3^, argon atmosphere. Heat treatment: Stress relieving at 200 °C followed by oven cooling.	Fatigue, porosity, modelling and simulation	Similar porosity percentage was found using optical microscopy and X-ray computed tomography techniques. Hot isostatic pressing post treatment resulted in reduction of strength, however, was comparable to that of the die-cast parts. Even smaller size pores present in the vicinity of the surface of the fatigue samples, significantly contributed to the decrease in fatigue life. This surface weakness effect was mitigated by the hot isostatic pressing post treatment.	SLM 250 HL	[116]
Siddique et al. (2017)	AlSi_12_	Laser power 350 W, scan speed 930 mm/s, hatch spacing 0.19 mm, layer thickness 50 µm, energy density 39.6 J/mm^3^, scanning strategy chessboard, hatch rotation 79°, argon atmosphere, base plate temperature 200 °C. Build orientation: Vertical. Heat treatment: Stress relieving at 200 °C for 2 h.	Fatigue, porosity, hardness, crack propagation testing	Stress relief post heat treatment at 240 °C caused an increase in porosity due to the growth of pores. At low stresses, the samples printed with base plate heating displayed higher fatigue performance compared to the samples printed without base plate heating. Samples printed without base plate heating consisted of higher porosity, which facilitated samples failing from cracking due to defects. Such an occurrence was significantly reduced in samples printed with base plate heating.	SLM 250 HL	[14]
Siddique et al. (2015)	AlSi_12_	Laser power of 400 W, energy density 20–39.6 J/mm^3^, chessboard scanning strategy, hatch rotation 79°, argon atmosphere, base plate temperature 200 °C. Build orientation: Vertical. Heat treatment: Stress relieving at 240 °C followed by oven cooling.	Tensile, surface roughness, residual stress analysis, fatigue	Base plate heating induces a coarser grain microstructure in the printed samples owing to a decrease in cooling rate. Tensile strength of the printed samples was four times that of sand-cast parts and two times that of die-cast parts. Significant reduction in residual stresses was observed in samples printed with base plate heating, which also reduced the scatter in fatigue data.	SLM 250 HL	[15]
Silvestri et al. (2020)	AlSi_10_Mg	EOS M400: Laser power 1000 W, laser spot diameter 90 µm, nitrogen atmosphere. SLM 280: Laser power 400–1000 W, laser spot diameter 80–225 µm, argon atmosphere. Renishaw AM400: Laser power 400 W, laser spot diameter 70 µm, argon atmosphere. Build orientation: 0°, 60°, 90°.	Tensile	The mechanical properties of the samples printed using different SLM machines were different, even though the best process parameters suggested by the equipment manufacturers were employed.	EOS M400 SLM 280 Renishaw AM400	[117]
Subbiah et al. (2020)	AlSi_10_Mg	Laser power 350 W, laser spot size 0.2 mm, scanning speed 730 mm/s, hatch spacing 0.12 mm, layer thickness 30 µm, stripe scanning strategy, inert atmosphere, base plate temperature 150 °C. Heat treatment: Solution treated at 550 °C for 2 h and water quenched.	Tensile, surface roughness, modelling and simulation	The microstructural studies revealed that the samples were stretched due to the exclusion of Si enriched cellular and dendritic network. Printed samples exhibited high tensile strength of 431 MPa.	SLM 280 HL	[118]
Sun et al. (2019)	AlSi_10_Mg	Laser power 400 W, laser spot size 100 µm, scanning speed 1000 mm/s, hatch spacing 0.165 mm, layer thickness 25 µm, chessboard scanning strategy, hatch rotation 67°. Heat Treatment: T2 treatment—annealed at 380 °C for 45 mins and air cooled, T6-like treatment—solution treated at 500 °C for 15 mins, quenched, and aged at 158 °C for 10 mins.	Tensile	A homogeneous distribution of spheroidised Si was observed in heat treated parts. It was suspected that Si experienced the necking effect under a tensile environment due to the large temperature gradient and α-Al erosion during the SLM process. The tensile strength of the as-built samples was better than the as-cast samples, however, the strength reduced with subsequent post heat treatment.	Renishaw AM 250	[119]
Suryawanshi et al. (2016)	AlSi_12_	Laser power 320 W, layer thickness 50 µm, hatch spacing 110 µm, hatch rotation 73°, scanning speed (1455 mm/s for volume and 1939 mm/s for contour), argon atmosphere, single melt and checkerboard scanning strategy. Heat treatment: solution treated at 573 K for 6 h.	Tensile, fracture toughness, fatigue crack growth	The fatigue crack growth threshold and unnotched fatigue strength of SLM alloys was inferior compared to cast alloys, which could be attributed to tensile residual stresses, shrinkage porosity, and un-melted particles. The printed samples exhibited enhanced toughness due to the presence of mesostructure Si. Toughness was found to be sensitive to crack orientation with respect to the build and scan orientations.	SLM 250 HL	[120]
Takata et al. (2017)	AlSi_10_Mg	Laser power 380 W, layer thickness 30 µm, hatch spacing 0.1 mm, hatch rotation 67°, argon atmosphere. Build orientation: Horizontal (X/Y), Vertical (Z). Heat treatment: Annealing at 300 °C for 2 h, or solution treatment at 530 °C for 6 h and water quenched.	Tensile	A fine dislocation substructure consisting of low angle boundaries was found within the α-Al grains. {001} texture along the Z direction was observed which was attributed to the preferential <001> grain growth of the α-Al phase during rapid solidification. The as-built samples exhibited a high tensile strength of approximately 480 MPa irrespective of the build orientation. In contrast, the ductility was direction-dependent, thereby resulting in the fracture preferentially occurring at the melt pool boundaries.	EOSINT M 280	[121]
Tang and Pistorius (2017)	AlSi_10_Mg	Laser power 370 W, scan speed 1300 mm/s, layer thickness 30 µm, laser spot diameter 100 µm, hatch spacing 0.16–0.22 mm, hatch rotation 67°. Build orientation: XY, Z. Heat treatment: Stress relieving at 573 K for 2 h.	Tensile, fracture morphology, porosity	The Z-oriented samples flow at a lower imposed stress than the XY-oriented samples. The maximum yield strength was noted for the XY-built samples, while maximum tensile strength was observed for the Z-built samples. Variation in hatch spacing results in porosity formation in the printed parts, which subsequently reduced the tensile performance.	EOS M280	[122]
Tang and Pistorius (2019)	AlSi_10_Mg	Laser power 370 W, scan speed 1300 mm/s, laser beam diameter 0.1 mm, layer thickness 30 µm, hatch spacing 0.16–0.22 mm, and the Build orientation: XY, Z. Heat treatment: Stress relieving at 573 K for 2 h.	Fatigue, porosity	A correlation between the crack-initiating pore on the fracture surface and fatigue life was established. The fatigue resistance was affected by the variation in hatch spacing and build orientation. XY-oriented samples have better fatigue performance, possibly due to anisotropy of pores, residual stress, and of melt-pool boundaries.	EOS M280	[123]
Tradowsky et al. (2016)	AlSi_10_Mg	Laser power 175 W, laser spot size 60 µm, scan speed 1025 mm/s, layer thickness 30 µm, scan spacing 97.5 µm, chess scanning strategy, island size 5.6 mm, argon atmosphere. Build orientation: Horizontal, Vertical. Heat treatment: Solution treated at 520 °C for 5 h, water quenched, and aged at 160 °C for 12 h followed by air cooling.	Tensile, porosity, modelling and simulation	Columnar grains were observed along building direction, with equiaxed grains found in-cross section. Irregular-shaped voids were observed in both the as-built and heat-treated conditions due to the formation of oxide layer. These pores were considerably reduced after hot isostatic pressing post treatment, however, the oxide layers remained. Heat treated and hot isostatic pressed samples had tensile properties exceeding that of the cast counterparts.	Concept Laser M2 cusing system	[124]
Trevisan et al. (2016)	A357	Laser power 195 W, spot size 100 µm, layer thickness 30 µm, scanning speed 1200 mm/s, hatch distance 0.1 mm, hatch rotation 67°, build plate temperature 100 °C. Heat treatment: Stress relieving at 300 °C for 2 h, T6 treatment involving solution treatment at 540 °C for 8 h, water quench, and ageing at 170 °C for 3 h.	Tensile, hardness	The tensile strength decreased after stress relief heat treatment. After T6 heat treatment, the microstructure became more isotropic, and the mechanical properties were comparable to that of the as-built condition.	EOSINT M270 Dual mode machine	[125]
Uzan et al. (2017)	AlSi_10_Mg	Laser power 400 W, beam diameter 100–150 µm, scanning speed 1000 mm/s, layer thickness 30 µm and 60 µm (before and after melting), hatch distance 200 µm, stripe scanning strategy, argon atmosphere, build plate temperature 35 °C. Build orientation: Z direction. Heat treatment: Stress relieved at 300 °C for 2 h.	Fatigue, tensile, fracture toughness, hardness	The fatigue resistance of the as-built samples was highest and that of the stress relieved and hot isostatic pressed samples was the lowest. The critical stress intensity factor can be estimated by the fracture surface morphology of the fatigue specimen.	EOSINT M-280	[126]
Vrana et al. (2016)	AlSi_10_Mg	Laser power 150–400 W, layer thickness 50 µm, scanning speed 1000–4000 mm/s, nitrogen atmosphere, build platform temperature 120 °C.	Impact testing	Lattice structures fabricated with scanning speed of 500 mm/s achieved 10× higher stiffness than those printed using 4000 mm/s for the same geometry of the unit-cell. Samples printed with a laser power of 350 W and scanning speed of 3000 mm/s achieve greater geometrical stability and had better accuracy.	SLM 280HL	[127]
Wang et al. (2019)	AlSi_10_Mg	Laser power 370 W, scan speed 1300 mm/s, hatch spacing 190 µm, layer thickness 30 µm, argon atmosphere. Build orientation: Vertical. Heat treatment: Stress relieved at 250 °C for 4 h.	Tensile, porosity, modelling and simulation	The deviation between reconstructed and as-designed models was less than 100 μm. The anisotropic properties of the printed parts were attributed to the non-uniform distribution of process-induced defects within the samples, which had a deleterious effect on the tensile strength. The presence of geometric defects significantly influenced the tensile strength and elongation.	EOS M290	[128]
Wang et al. (2018)	AlSi_10_Mg	Laser power 400 W, laser spot size 76 µm, scanning speed 1000 mm/s, hatch distance 175 µm, layer thickness 25 µm, chessboard scanning strategy, island size 5 mm, argon atmosphere. Heat treatment: T6 solution treated at 535 °C for 7–15 mins and aged at 158 °C for 10 h.	Tensile, bending, hardness	A decrease of about 20% in hardness and tensile strength whereas an increase of about 155% in elongation was reported for the heat-treated samples. An increase of about 123% in fracture deflection and a decrease of about 6% in bending strength was also found for the heat-treated samples.	Renishaw AM 250	[129]
Wang et al. (2018)	AlSi_10_Mg	Laser power 400 W, scanning speed 1000 mm/s, hatch spacing 175 µm, layer thickness 25 µm, chessboard scanning strategy, hatch rotation 67°. Build orientation: Parallel, Normal. Heat Treatment: T2 stress relieving treatment at 380 °C for 45 mins followed by air cooling.	Tensile, bending, hardness	The hardness, tensile strength and bending strength decrease by about 50% after T2 heat treatment The precipitates in the molten pool boundaries dissolve in the matrix after heat treatment.	Renishaw AM 250 system	[130]
Wang et al. (2014)	AlSi_12_	Laser power 200 W, beam diameter 35 µm, scanning speed 375–2000 mm/s, hatch spacing 0.15 mm, layer thickness 50 µm, stripe scanning strategy, hatch rotation 90°, inert atmosphere.	Tensile, hardness	Insignificant differences in density and hardness was observed when the samples were printed in either argon, nitrogen, or helium chamber atmospheres. The samples showed superior performance; 1.5 times yield strength, 20% higher tensile strength, and twice elongation, compared to conventionally produced material.	Realizer SLM 100	[131]
Wei et al. (2017)	AlSi_10_Mg	Laser spot size 70 µm, laser power 150–180 W, scanning speed 600–1400 mm/s, layer thickness 40 µm, hatch spacing 50–70 µm, random scanning strategy, argon atmosphere.	Tensile, fracture morphology	A linear energy density of 1.5–1.875 J/cm was reported to yield continuous single-track depositions. The level of porosity was significantly influenced by the variation in hatch spacing. The pores and un-melted particles cause reduction in tensile strength and strain.	Self-developed SLM 150 equipment	[132]
Wu et al. (2016)	AlSi_10_Mg	Laser power 175 W, scanning speed 1030 mm/s, hatch spacing 0.65 mm, island size 6 mm.	Tensile, in-situ compression testing	Long cell-like structures are formed in the printed samples owing to the high cooling rate of the process. The Al within the eutectic grows epitaxially on the pre-existing Al cell resulting in further epitaxial growth of the Al cell formed above the eutectic. Si precipitates present across the Al-matrix inhibits dislocation motion within the large Al grains.	Concept Laser M2 system	[133]
Yan et al. (2014)	AlSi_10_Mg	Laser beam diameter 100 µm, laser power of 400 W, layer thickness 30 µm, argon atmosphere.	Compression, computed tomography	Lattices with cell sizes 3–6.5 mm were very thin for a low volume fraction of 5%, which tended to break during printing. The printed samples showed good geometric agreement with the CAD model. The compressive modulus and strength of the printed samples directly proportional to the volume fraction of the lattice cells and inversely proportional to the unit cell sizes.	EOSINT M280	[134]
Yang et al. (2018)	A357	Laser power 750 W, scanning speed 1100 mm/s, hatch rotation 90°, build plate temperature 35 °C. Build orientation: Horizontal, Vertical. Heat Treatment: Directly aged at 160 °C for 8 h; Stress-relieved at 300 °C for 2 h; Stress-relieved and solution treated at 543 °C for 1–8 h, quenched and aged at 160 °C for 8 h.	Tensile, porosity	Anisotropy in ductility between horizontally and vertically built samples decrease with solution heat treatments carried out for longer periods of time. Negligible coarsening of the columnar grains was observed for all heat treatment conditions. The stress relieved sample displayed highest elongation to fracture, primarily owing to the break-up of the Si network into fully dispersed Si particles.	Concept Laser X-line 1000 machine	[26]
Zaretsky et al. (2017)	AlSi_10_Mg	Build orientation: XY, Z. Heat treatment: T5 stress relieving at 300 °C for 2 h.	Dynamic and quasi-static tensile	Fracture of the printed samples at different tensile rates reflect a change in fracture mode from rate-independent ductile mode to rate-dependent brittle mode. At high plastic strain rate range, the yield strength of the printed samples was strain rate sensitive.	EOSINT M280 system	[135]
Zhang et al. (2018)	AlSi_10_Mg	Laser power 490 W, scanning speed 2000 mm/s, layer thickness 40 µm, hatch spacing 0.1 mm, hatch rotation 90°, argon atmosphere. Heat treatment: Stress relieved at 300 °C for 2 h; Solution treated at 530 °C for h, water quenched, and aged at 170 °C for 12 h.	Fatigue, tensile	Heat treatment reduces fatigue property due to the coarsening of Si precipitates. For most of the fractured surfaces, the crack initiation sites indicated a presence of surface or subsurface defects. The as-built samples display higher fatigue property compared to heat treated samples.	Self-developed SLM system (LSNF-2)	[136]
Zhou et al. (2018)	AlSi_10_Mg	Laser power 400 W, scanning speed 1300 mm/s, layer thickness 30 µm, hatch spacing 0.19 mm. Heat treatment: T6 solution treated at 520 °C for 0.5–4 h, water quenched, and aged at 160 °C for 1–24 h.	Hardness	Eutectic structure up to 4 μm in size containing tiny needle-like semi-coherent Si particles were observed within the Al cells. Embedded spherical nanoscale Si particles and segregated Mg and Fe were observed along the cell and grain boundaries of primary Al. The π-Al_8_Si_6_Mg_3_Fe precipitates were also identified along the boundaries. Peak hardening was observed after ageing at 160 °C for 6–10 h, and remained relatively unchanged up to 24 h treatment time.	EOS M280	[137]
Zhou et al. (2019)	AlSi_10_Mg	Laser power 300 W, scanning speed 800 mm/s, laser spot size 80 µm, hatch spacing 0.13 mm, layer thickness 30 µm, volumetric energy density 62.5 J/mm^3^, argon atmosphere, build plate temperature 150 °C. Heat treatment: Stress relieved at 300 °C for 2 h and water quenched; solution treated 535 °C for 1 h, water quenched, and aged at 190 °C for 10 h.	Tensile	Stress relieving post heat treatment was effective for eliminating residual stresses. The printed samples exhibited tensile strength of 273.2 MPa and plasticity of 15.3%.	SLM 280 HL	[138]

**Table 2 materials-13-04301-t002:** Overview of research focus areas into four categories.

Aluminium Alloy	Process Parameter Optimisation	Build Orientation	Heat Treatment	Numerical Studies
**AlSi_10_Mg**	[38,73,76,77,79,80,88,108]	[10,12,31,33,37,40,41,42,43,49,50,59,61,66,68,71,73,75,81,89,93,94,95,110,113,117,122,126,135,144]	[10,11,27,28,50,57,58,59,63,65,67,70,80,90,91,92,94,95,110,111,118,119,121,124,126,129,130,136,137,138,145,146]	[29,52,69,83,84,87,96,97,114,118,128]
**AlSi_12_**	[34]	[99,104,120]	[15,16,17,18,82,100,101,106,120]	-
**A356**	-	-	[20]	-
**A357**	-	[25,26,36,48,51]	[24,26,32,48,125]	[64]

**Table 3 materials-13-04301-t003:** List of studies evaluating various mechanical properties of SLM-printed Aluminium alloys.

Property	AlSi_10_Mg	AlSi_12_	A356	A357
**Microhardness**	[50,52,53,58,66,68,70,73,76,77,79,86,92,107,111,112,126,129,130,137,145]	[45,131]	-	[125]
**Nanohardness**	[11,12,28,49,54]	-	-	-
**Tensile**	[4,10,11,12,30,33,42,43,44,49,53,58,63,65,67,68,70,71,72,73,76,78,80,81,85,88,89,90,92,98,103,107,109,112,113,117,118,119,121,122,124,126,128,129,130,132,136,138,144,146]	[15,17,18,55,74,101,104,115,120,131]	[20]	[24,25,26,32,36,48,51,64,125]
**Compressive**	[11,12,38,102,134]	-	-	-
**Fatigue**	[27,35,37,40,92,93,98,123,126,136,144]	[14,115,116,120]	-	[36]
**Fracture Toughness**	[40,43,49,53,59,66,73,89,90,92,112,118,122,126,132,144]	[55,120]	-	-
**Dynamic**	[61,62,89,95]	[99,100,147]	-	-
**Creep Resistance**	[108]	-	-	-
**Impact**	[103,107,110,127,135]	-	-	-
**Wear**	[47,86]	[106]	-	-
**Flexural/Bending**	[129,130]	[105]	-	-
**Shear**	-	-	-	[51]

**Table 4 materials-13-04301-t004:** Mechanical properties of SLM-printed AlSi_10_Mg.

**Effect of Build Orientation:**					
**Build Orientation ^a^**	**Hardness ^b^**	**YS (MPa)**	**UTS (MPa)**	**Strain (%)**	**Reference**
H, V	109.7 (max)	200	360	2	[12]
H, I, V	-	241.2 (H)	379.6 (H)	8.1 (H)	[33]
		239.1 (I)	367.8 (I)	5.7 (I)	
		236.8 (V)	351.8 (V)	8.3 (V)	
H, V	-	-	338 (H)	-	[43]
		-	385 (V)	-	
H, I, V	130.6 (max)	206.74 (0°,5°)	366.43 (0°,5°)	-	[67]
		241.15 (0°, 5°)	399.10 (0°, 5°)	-	
		222.83 (0°, 85°)	360.27 (0°, 85°)	-	
		188.15 (45°, 0°)	330.11 (45°, 0°)	-	
		179.71 (45°, 5°)	314.32 (45°, 5°)	-	
		208.57 (90°, 45°)	357.49 (90°, 45°)	-	
H, V	-	-	340 (H)	-	[81]
		-	350 (V)	-	
H, V	94	170 (H) (max)	277 (H) (max)	-	[112]
		170 (V) (max)	267 (V) (max)	-	
H, V	-	195 (H)	338 (H)	11.1 (H)	[113]
		187 (V)	331 (V)	11.5 (V)	
H, V	-	187 (H)	284 (H)	-	[122]
		191 (V)	274 (V)	-	
H, I, V	-	-	250 (H)	-	[68]
		-	190 (I)	-	
		-	220 (V)	-	
H, V	127 (H)	-	391 (H)	-	[4]
	86 (V)	-	396 (V)	-	
**Effect of Heat Treatment:**					
**AB/HT ^c^**	**Hardness**	**YS (MPa)**	**UTS (MPa)**	**Strain (%)**	**Reference**
AB	-	300	455	0.08	[44]
AB	-	224.3	349.5	-	[53]
AB	-	-	380	-	[56]
AB	-	265	375	-	[124]
AB	-	-	360	6	[132]
AB	1.52 GPa	218	312	1.80	[49]
AB	-	240	360	-	[78]
AB	118 (Max)	186	354	-	[88]
AB	2.098 GPa	270	446	8.09	[138]
AB	-	248 (H)	386 (H)	8.6 (H)	[42]
		228 (V)	412 (V)	7 (V)	
HT-T6		243 (H)	323 (H)	15.3 (H)	
		223 (V)	302 (V)	16 (V)	
HT-T5		321 (H)	471 (H)	8.6 (H)	
		292 (V)	493 (V)	6 (V)	
AB	140.7	255	377	-	[58]
SR		158	256	-	
HT-T6		210	284	-	
AB	123	200	400	-	[63]
HT	51	100	150	-	
AB	127	-	307	-	[70]
HT		-	382 (max)	-	
AB	136	-	396	-	[76]
HT	152	-	399	-	
AB	132	-	434	-	[80]
HT	95	-	168	-	
AB	-	-	434	-	[10]
HT		-	200	-	
AB	-	-	470	-	[92]
HT		-	160	-	
HT-T2	53 HB (max)	105	171	-	[130]
HT-T6	100	239	333	4.50	[11]
HT	143.33	-	499	-	[72]
**Quasi-static and Dynamic Compressive Properties:**					
**QS/D ^d^**	**UCS (MPa)**		**YS (MPa)**	**Strain (%)**	**Reference**
QS-C	560		350	10	[38]
D-C	590		-	0.3	[94]
D-C	700		-	0.13	[61]
**Fatigue and Bending Properties:**					
**Other tests ^e^**	**Fatigue Strength/No. of cycles**		**Fracture Toughness (MPa.m^0.5^)**		**Reference**
AF, FT	3 × 10^7^		99 (max)		[27]
RB	5 × 10^7^		-		[40]
AF, FT	3 × 10^7^		94		[91]
AF, FT	10^7^ at 125 MPa		37.4		[126]
AF, FT	120 MPa		27		[136]
AF	100 MPa (max)		-		[123]
FT (at build orientations)	-		59.06 (0°, 5°)		[66]
	-		51.60 (0°, 5°)		
	-		58.03 (0°, 85°)		
	-		55.79 (45°, 0°)		
	-		50.76 (45°, 5°)		
	-		40.63 (90°, 45°)		

**Key:**^a^—H (Horizontal), I (Inclined at 45°), V (Vertical); ^b^—Measurements reported in HV (Vickers Hardness), unless otherwise specified differently; ^c^—AB (As-built), SR (Stress Relief), HT (Heat-treated); ^d^—QS (Quasi-static testing), D-T (Dynamic-Tension testing), D-C (Dynamic-Compression testing); ^e^—AF (Axial Fatigue), RB (Rotating Bending), FT (Fracture Toughness).

**Table 5 materials-13-04301-t005:** Mechanical properties of SLM-printed AlSi_12_.

**Effect of Build Orientations:**					
**Build Orientation ^a^**	**Hardness ^b^**	**YS (MPa)**	**UTS (MPa)**	**Strain (%)**	**Reference**
H, V	-	270 (H)	325 (H)	4.4 (H)	[120]
		274 (V)	296 (V)	2.2 (V)	
H, I, V	-	227 (H)	260 (H)	2.0 (H)	[104]
		263 (I)	367 (I)	4.5 (I)	
		224 (V)	398 (V)	5.0 (V)	
**Effect of Heat Treatment:**					
**AB/HT ^c^**	**Hardness ^b^**	**YS (MPa)**	**UTS (MPa)**	**Strain (%)**	**Reference**
AB	-	290	460	-	[101]
AB	-	263	365	-	[104]
AB	-	201	361	4	[116]
AB	115	224 (max)	368 (max)	4.8	[131]
AB	-	240	325	-	[18]
HT		138	207	-	
AB	-	220	418	3.9	[15]
HT		218	372	3.4	
HT	-	102	425	12	[55]
**Dynamic Compressive Properties:**					
**QS/D ^d^**	**UTS (MPa)**	**UCS (MPa)**	**YS (MPa)**	**Strain (%)**	**Reference**
D-C at RT	-	550	400	0.18	[99]
D-C at 200 °C	-	490	270	0.18	[147]
**Fatigue and Fracture Toughness:**					
**Other tests ^e^**	**Fatigue Strength/No. of cycles**		**Fracture Toughness (MPa.m^0.5^)**		**Reference**
AF	10^9^ at 60.5 ± 4.7 MPa		-		[15]
FT	-		19.7		[120]

**Key:**^a^—H (Horizontal), I (Inclined at 45°), V (Vertical); ^b^—Measurements reported in HV (Vickers Hardness), unless otherwise specified differently; ^c^—AB (As-built), HT (Heat-treated); ^d^—QS (Quasi-static testing), D-C (Dynamic-Compression testing); ^e^—AF (Axial Fatigue), FT (Fracture Toughness).

**Table 6 materials-13-04301-t006:** Mechanical properties of SLM-printed A356 and A357.

**Effect of Build Orientation:**					
**Build Orientation ^a^**	**Hardness ^b^**	**YS (MPa)**	**UTS (MPa)**	**Strain (%)**	**Reference**
H, V	-	257 (H)	398 (H)	4.4	[48]
		216 (V)	400 (V)	2.2	
H, I, V	-	184 (H)	284 (H)	-	[51]
	-	195 (I)	298 (I)	-	
	-	192 (V)	305 (V)	-	
**Effect of Heat Treatment:**					
**AB/HT ^c^**	**Hardness ^b^**	**YS (MPa)**	**UTS (MPa)**	**Strain (%)**	**Reference**
AB *	-	250	400	-	[20]
HT–T5 *	-	125	200	-	
AB	-	279	426	-	[25]
HT		205	307	-	
AB	-	279	426	-	[24]
SR	-	165	240	-	
AB	-	225	375	-	[26]
SR	-	125	220	-	
HT–T6	138	200	400	5.5	[32]
**Fatigue and Fracture Toughness:**					
**Other tests ^d^**	**Fatigue Strength/No. of cycles**		**Fracture Toughness (MPa.m^0.5^)**		**Reference**
AF	2 × 10^6^ at 60 MPa		-		[36]

**Key:**^a^—H (Horizontal), I (Inclined at 45°), V (Vertical); ^b^—Measurements reported in HV (Vickers Hardness), unless otherwise specified differently; ^c^—AB (As-built), HT (Heat-treated), SR (Stress Relief); ^d^—AF (Axial Fatigue), FT (Fracture Toughness); *—A356 alloy.

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
