# Peer review of "Mechanical Properties of SLM-Printed Aluminium Alloys: A Review"

_materials, 2020, doi:10.3390/ma13194301_

Round 1

Reviewer 1 Report

To improve the manuscript, please provide additional information, as follows:

  • For Fig. 1 (reference [2]) should be obtained the Rights and Permission from the publisher (Copyright © 2014, Springer Nature); see the link Reprints and Permissions from https://link.springer.com/article/10.1007/s11665-014-0958-z
  • the abbreviations should be defined in the first place where are used (i.e. SLM was abbreviated on page 1 and page 4, also Al-Si alloy was used on page 4, line 85, but was abbreviated from Aluminium-Silicon (Al-Si) alloys on page 4, line 87);
  • on the Discussion section, the comments should be carried out by specifying the values of the properties of the SLMed parts, i.e. on page 5, line 140, at rapid cooling, the cooling rate should be provided, on page 5, lines 154-155, at "(a) optimizing powder characteristics (b) optimizing SLM process parameters for attaining nearly 100% relative density" should be mentioned the type of powder characteristics, as well as the SLM process parameters;
  • Table 1 should be improved since the process conditions (build orientation, heat treatment, and process optimisation), types of mechanical properties and type of SLM machine used are too general described; more information should be specified on the process parameters (i.e. at heat treatment should be mentioned atmosphere, heating temperature and time, heating rate, cooling rate, at SLM process should be mentioned the main process parameters such as laser power, scan speed, layer thickness, and scan strategy); the values of the properties should be given (quantified) and the main findings should be specified more clear to highlight how the properties of the SLMed parts were improved in comparison with the parts manufactured by conventional processes; also discussion on the shape and size of the SLMed parts should be carried out since these characteristics influence the other properties; at the SLM machine should be specified at least laser type and power and the spot size as are given for the reference [53];
  • in Table 1, it was claimed that "The fatigue strength at 2 × 10^6 cycles was statistically analysed according to ISO 12107: 2012" but the other properties from the literature reports are also analyzed by the standards in force that are not mentioned in Table 1; in this respect even delete the mentioned standard or add the other standards;
  • in Table 1, Kempen et al. (2011), in "Average roughness values of 20 μm Pa value were measured" delete "Pa";
  • in the Conclusion section, it was claimed that "Moreover, no published studies have been found on high strain rate dynamic behaviour of SLM processed Aluminium alloys tested at elevated operating temperatures" but the reference [153] refers to "High strain rate behaviour at high temperature of AlSi12 parts produced by selective laser melting"; please clarify this issue;
  • check the references since the following references are incomplete, i.e. [1], [11], [61-63], [68], [72], [73], [79], [81], [85], [87], [88], [92], [98], [102], [103], [112], [114], [124], [129], [131], [146], [148], [151], [153], [159], [160], [162].

Author Response

1.For Fig. 1 (reference [2]) should be obtained the Rights and Permission from the publisher (Copyright © 2014, Springer Nature); see the link Reprints and Permissions from https://link.springer.com/article/10.1007/s11665-014-0958-z

Response: The permission for reuse of Figure 1 has been obtained and the RightsLink License is attached at the end of the revised manuscript.

2.The abbreviations should be defined in the first place where are used (i.e. SLM was abbreviated on page 1 and page 4, also Al-Si alloy was used on page 4, line 85, but was abbreviated from Aluminium-Silicon (Al-Si) alloys on page 4, line 87).

Response: This has been corrected.

3.On the Discussion section, the comments should be carried out by specifying the values of the properties of the SLMed parts, i.e. on page 5, line 140, at rapid cooling, the cooling rate should be provided, on page 5, lines 154-155, at "(a) optimizing powder characteristics (b) optimizing SLM process parameters for attaining nearly 100% relative density" should be mentioned the type of powder characteristics, as well as the SLM process parameters.

Response: The details of the cooling rate and the optimum set of process parameters have been mentioned in Table 1 of the revised manuscript.

4.Table 1 should be improved since the process conditions (build orientation, heat treatment, and process optimisation), types of mechanical properties and type of SLM machine used are too general described; more information should be specified on the process parameters (i.e. at heat treatment should be mentioned atmosphere, heating temperature and time, heating rate, cooling rate, at SLM process should be mentioned the main process parameters such as laser power, scan speed, layer thickness, and scan strategy); the values of the properties should be given (quantified) and the main findings should be specified more clear to highlight how the properties of the SLMed parts were improved in comparison with the parts manufactured by conventional processes; also discussion on the shape and size of the SLMed parts should be carried out since these characteristics influence the other properties; at the SLM machine should be specified at least laser type and power and the spot size as are given for the reference [53].

Response: As per the suggestion of the Reviewer the contents of Table 1 have been thoroughly revised. We haved added information regarding the process parameters, build orientations, heat treatments, mechanical properties, and main findings for each study referenced in Table 1.

5.In Table 1, it was claimed that "The fatigue strength at 2 × 10^6 cycles was statistically analysed according to ISO 12107: 2012" but the other properties from the literature reports are also analyzed by the standards in force that are not mentioned in Table 1; in this respect even delete the mentioned standard or add the other standards.

Response: The mentioned standard has been deleted from Table 1 in the revised manuscript.

6.In Table 1, Kempen et al. (2011), in "Average roughness values of 20 μm Pa value were measured" delete "Pa".

Response: This has been corrected.

7.In the Conclusion section, it was claimed that "Moreover, no published studies have been found on high strain rate dynamic behaviour of SLM processed Aluminium alloys tested at elevated operating temperatures" but the reference [153] refers to "High strain rate behaviour at high temperature of AlSi12 parts produced by selective laser melting"; please clarify this issue.

Response: The Conclusion section has been revised to incorporate changes suggested by the Reviewers. This particular phrase has been removed from the revised manuscript.

8.Check the references since the following references are incomplete, i.e. [1], [11], [61-63], [68], [72], [73], [79], [81], [85], [87], [88], [92], [98], [102], [103], [112], [114], [124], [129], [131], [146], [148], [151], [153], [159], [160], [162].

Response: All the references have been checked and are completed in the revised manuscript.

Reviewer 2 Report

The manuscript titled: ‘’Mechanical properties of SLM-printed Aluminium alloys: A Review’’ systematically and comprehensive describes various studies on mechanical properties of SLM processed aluminium alloys – what was done in the last decade. The main objective of this review article is to evaluate the research directions and research gaps for different types of mechanical properties of differently SLM performed aluminium alloys. This topic has immense importance due to increased use of SLM production of different aluminium alloys.  

However, the paper is interesting but it needs to be improved. Many sentences and thoughts are mentioned too many times. The whole text needs unification!

The manuscript (the whole text) needs to be unified for some parts like:

  • Write aluminium alloy always with small letters, not once with capital letters and once with lowercase letters.
  • You repeat exactly the same sentences or use sentences with the same meaning too many times. It is being disturbing during the reading.

1st Introduction:

  • Line 65: use SLM as abbreviation not with words once it has been explained (as well as in lines 88, 95…133, 139, 217, etc. - unify that in the whole text)
  • 2 has low resolution quality. Because the writings is so small the resolution should be improved or make scheme better readable. What is LFF system? Explain abbreviation.

2nd Chapter:

  • Lines 95-96 and 102-103: There is the same sentence. Omit the second one! It is no needed to be repeated so many times.
  • Lines 104-109 are being repeated with exact the same words from lines 378-382. Rewrite the second repetition.
  • Line 108: There is no evidence nor references regarding the cryogenic usage of the material. Write more about it.
  • Line 111: sentence. ‘’Regarding AlSi12 in SLM, it is still evolving.’’ Is already being mention. Omit this one!
  • Lines 160-165 are exactly the same as lines 371-376. Omit the second repetition.
  • Table 1: there are some abbreviations used (FEA, AA, SHT, Et, PD, DMLS, HPDC, SHT, etc.) and are not explained anywhere in the text. Because there are so many abbreviations in the manuscript I suggest to make a table of all abbreviations and their explanations used in the paper.

Some units in table 1 are not written correctly (typographical error) - °C at ref. 92 and ref. 116, Al6061 should be written AA 6061 at ref. 95, use correct writing for phases – ref. 99…

3rd Chapter:

  • Line 240 has one ‘that’ too much.
  • Lines 253-254: sentence ‘’AlSi10Mg is one of the most common aluminium alloys and finds wide applications in various industries of 3D printing applications such as automotive and aerospace.’’ Is written too many times in the text. Rewrite it!
  • Lines 261-262. Why is ‘’Fatigue and Fracture Toughness’’ written with capitals?
  • Table 4: From the results between AB and HT in table 4 can be seen the deviation between different authors. Did you investigate the reasons for those deviations (mostly higher UTS value for AB, but not with all authors)? What is the reason for that?
  • Line 349: use W for watts.
  • Lines 413-414: TNM is explained after first usage of abbreviation. Make it vice-verso.
  • Line 441: use expression plate heating instead of plating heating.
  • Lines 521-522: sentence ‘’Furthermore, the single melt sample was found to have more porosity than checkerboard scanning strategy.’’ In this sentence you are talking about and comparing two different things….melting and scanning strategy. Checkerboard scanning strategy can also have single melting. Revise the sentence to get the correct meaning.
  • Line 559: ‘’ the intermetallic phases and Mg2Si phases’’ what is the difference here and what ‘other’ intermetallic phases you meant beside Mg2Si?
  • Line 566: use ‘using SLM’ instead of ‘on SLM’.
  • Line 575: use ‘phases’ instead of ‘agglomerates’ (if the meaning is staying the same).
  • Line 639: use correct writing of units
  • Line 656: use interspace between ‘’AA 6061’’ and use ‘with SLM’ instead of ‘in SLM’

4th Chapter:

  • Rewrite the first sentence, because it is hard to read.
  • There is no highlights in the Summary to be pointed out. Make more outstanding claims and conclusions of the research.
  • Did you trace anything about the repeatability of the SLM process and comparison of the used similar/same process parameters on different SLM machines?

References:

Many research works from the last decade have been reviewed and evaluated in this work. Well done!

  • References have some minor typographical errors (ref. 49 – capitals for writing chemical elements…)

Author Response

Responses and corrections in accordance with the comments of Reviewer #2:

The manuscript titled: ‘’Mechanical properties of SLM-printed Aluminium alloys: A Review’’ systematically and comprehensive describes various studies on mechanical properties of SLM processed aluminium alloys – what was done in the last decade. The main objective of this review article is to evaluate the research directions and research gaps for different types of mechanical properties of differently SLM performed aluminium alloys. This topic has immense importance due to increased use of SLM production of different aluminium alloys. However, the paper is interesting but it needs to be improved. Many sentences and thoughts are mentioned too many times. The whole text needs unification!

Response: The authors are pleased that the Reviewer found our work to be insightful. We appreciate the Reviewer to have pointed out the shortcomings in our manuscript and we have addressed all of the Reviewer’s concerns to the best of our ability.

  1. The manuscript (the whole text) needs to be unified for some parts like:
  • Write aluminium alloy always with small letters, not once with capital letters and once with lowercase letters.
  • You repeat exactly the same sentences or use sentences with the same meaning too many times. It is being disturbing during the reading.

Response: The manuscript has been thoroughly revised and unnecessary repetitions and typographical errors have been corrected.

  1. Line 65: use SLM as abbreviation not with words once it has been explained (as well as in lines 88, 95…133, 139, 217, etc. - unify that in the whole text).

Response: The abbreviation of SLM is used elsewhere in the manuscript after its first mention.

  1. Figure 2 has low resolution quality. Because the writings is so small the resolution should be improved or make scheme better readable. What is LFF system? Explain abbreviation.

Response: The resolution of Figure 2 has been increased to 300 dpi. The full form of LFF is Laser Freeform Fabrication. It covers all laser-based additive manufacturing systems. This information has been added to the revised manuscript.

  1. Lines 95-96 and 102-103: There is the same sentence. Omit the second one! It is no needed to be repeated so many times.

Response: These sentences have been deleted, and replaced with the below sentence.

“Although significant amount of works are published on AlSi10Mg, the research work on AlSi12 and other Al-Si alloys including A356 and A357 is still evolving [11-26].”

  1. Lines 104-109 are being repeated with exact the same words from lines 378-382. Rewrite the second repetition.

Response: This paragraph has been deleted.

  1. Line 108: There is no evidence nor references regarding the cryogenic usage of the material. Write more about it.

Response: This sentence has been moved to Line 306-307 of the revised manuscript and a reference is provided that reports on the cryogenic application of AlSi12 aluminium alloy.

  1. Line 111: sentence. ‘’Regarding AlSi12 in SLM, it is still evolving.’’ Is already being mention. Omit this one!

Response: This sentence has been deleted.

  1. Lines 160-165 are exactly the same as lines 371-376. Omit the second repetition.

Response: Lines 160-165 have been deleted.

  1. Table 1: there are some abbreviations used (FEA, AA, SHT, Et, PD, DMLS, HPDC, SHT, etc.) and are not explained anywhere in the text. Because there are so many abbreviations in the manuscript I suggest to make a table of all abbreviations and their explanations used in the paper.

Response: Table 1 has been thoroughly revised. The abbreviations have been replaced with full forms throughout the Table.

  1. Some units in table 1 are not written correctly (typographical error) - °C at ref. 92 and ref. 116, Al6061 should be written AA 6061 at ref. 95, use correct writing for phases – ref. 99…

Response: Any reference to Al6061 has been changed to AA 6061 throughout the revised manuscript.

  1. Line 240 has one ‘that’ too much.

Response: This has been corrected in Line 162 of the revised manuscript.

  1. Lines 253-254: sentence ‘’AlSi10Mg is one of the most common aluminium alloys and finds wide applications in various industries of 3D printing applications such as automotive and aerospace.’’ Is written too many times in the text. Rewrite it!

Response: The unnecessary repetition of these sentences has been deleted.

  1. Lines 261-262. Why is ‘’Fatigue and Fracture Toughness’’ written with capitals?

Response: This has been corrected.

  1. Table 4: From the results between AB and HT in table 4 can be seen the deviation between different authors. Did you investigate the reasons for those deviations (mostly higher UTS value for AB, but not with all authors)? What is the reason for that?

Response: There is no detailed research that can relate the differences in the mechanical performance of samples printed using different SLM machines. The authors believe that the potential reason for deviation in mechanical data between different authors is primarily due to variations in SLM machines and process parameters employed. This has been mentioned in the Conclusion section of the revised manuscript as a future research recommendation.

  1. Line 349: use W for watts.

Response: This has been corrected in Line 274 of the revised manuscript.

  1. Lines 413-414: TNM is explained after first usage of abbreviation. Make it vice-verso.

Response: This has been corrected in Line 338 of the revised manuscript.

  1. Line 441: use expression plate heating instead of plating heating.

Response: This has been corrected.

  1. Lines 521-522: sentence ‘’Furthermore, the single melt sample was found to have more porosity than checkerboard scanning strategy.’’ In this sentence you are talking about and comparing two different things….melting and scanning strategy. Checkerboard scanning strategy can also have single melting. Revise the sentence to get the correct meaning.

Response: This sentence has been deleted.

  1. Line 559: ‘’ the intermetallic phases and Mg2Si phases’’ what is the difference here and what ‘other’ intermetallic phases you meant beside Mg2Si?

Response: This has been corrected.

  1. Line 566: use ‘using SLM’ instead of ‘on SLM’.

Response: This phrase has been corrected.

  1. Line 575: use ‘phases’ instead of ‘agglomerates’ (if the meaning is staying the same).

Response: The reference article specifies the Al3Ni precipitates to be agglomerates. Therefore, this has been kept as mentioned previously.

  1. Line 639: use correct writing of units.

Response: This has been corrected in Line 562 of the revised manuscript.

  1. Line 656: use interspace between ‘’AA 6061’’ and use ‘with SLM’ instead of ‘in SLM’.

Response: This has been corrected in Lines 576-579 of the revised manuscript.

  1. 4thChapter: Rewrite the first sentence, because it is hard to read.

Response: The first sentence of the Summary section has been rewritten as below.

“Although aluminium can be with Zn, Cu, Mg, Mn and Si to produce age-hardening alloys, casting alloys and work-hardening alloys, the area of focus in this review was the Al-Si alloys processed by selective laser melting (SLM) metal additive manufacturing technique and their mechanical properties.”

  1. There is no highlights in the Summary to be pointed out. Make more outstanding claims and conclusions of the research.

Response: The Summary section has been revised as per the suggestion of the Reviewers.

  1. Did you trace anything about the repeatability of the SLM process and comparison of the used similar/same process parameters on different SLM machines?

Response: We did find that there was poor repeatability of the mechanical properties of SLM printed parts of the same aluminium alloy using the same machine but from different researchers. It was quite difficult to understand the effect of process parameters from different SLM machines as there was missing process parameter information in the literature. We have added a paragraph in the Summary section to recommend future research in investigating the repeatability of the SLM process.

  1. Many research works from the last decade have been reviewed and evaluated in this work. Well done! References have some minor typographical errors (ref. 49 – capitals for writing chemical elements…)

Response: The authors are pleased with the appreciation of the Reviewer. The typographical errors have been corrected.

Reviewer 3 Report

The article has a high scientific interest, since selective laser fusion is a process with a great industrial future.

The methods and materials are very well defined. Metallic additives, with powder bed fusion, have great results, which increases their industrial interest.

The results in terms of mechanical properties are excellent. The resistance to both static and dynamic loads is very high.

Its application in the biomedical, military and aerospace industry will increase greatly in the coming years.

Within future lines, it would be interesting to study the influence of heat treatments.

It is of sufficient quality to be published. The conclusions should be reviewed and complete.

Author Response

The article has a high scientific interest, since selective laser fusion is a process with a great industrial future. The methods and materials are very well defined. Metallic additives, with powder bed fusion, have great results, which increases their industrial interest. The results in terms of mechanical properties are excellent. The resistance to both static and dynamic loads is very high. Its application in the biomedical, military and aerospace industry will increase greatly in the coming years.

Within future lines, it would be interesting to study the influence of heat treatments.

It is of sufficient quality to be published. The conclusions should be reviewed and complete.

Response: The authors are pleased that the Reviewer found our work to be insightful. We appreciate the Reviewer to have pointed out the shortcomings in our manuscript, especially in the Conclusions and the Future Works sections and we have addressed these issues to the best of our ability.

Round 2

Reviewer 1 Report

The revised manuscript is recommended for publishing in Materials since the authors answered satisfactorily to the raised questions.